# Tabular Deep Learning when $d \gg n$ by Using an Auxiliary Knowledge Graph

## Abstract

Machine learning models exhibit strong performance on datasets with abundant labeled samples. However, for tabular datasets with extremely high $d$-dimensional features but limited $n$ samples (i.e. $d \gg n$), machine learning models struggle to achieve strong performance. Here, our key insight is that even in tabular datasets with limited labeled data, input features often represent real-world entities about which there is abundant prior information which can be structured as an auxiliary knowledge graph (KG). For example, in a tabular medical dataset where every input feature is the amount of a gene in a patient's tumor and the label is the patient's survival, there is an auxiliary knowledge graph connecting gene names with drug, disease, and human anatomy nodes. We therefore propose PLATO, a machine learning model for tabular data with $d \gg n$ and an auxiliary KG with input features as nodes. PLATO uses a multilayer perceptron (MLP) to predict the output labels from the tabular data and the auxiliary KG with two methodological components. First, PLATO predicts the parameters in the first layer of the MLP from the auxiliary KG. PLATO thereby reduces the number of trainable parameters in the MLP and integrates auxiliary information about the input features. Second, PLATO predicts different parameters in the first layer of the MLP for every input sample, thereby increasing the MLP's representational capacity by allowing it to use different prior information for every input sample. Across 10 state-of-the-art baselines and 6 $d \gg n$ datasets, PLATO exceeds or matches the prior state-of-the-art, achieving performance improvements of up to 10.19%. Overall, PLATO uses an auxiliary KG about input features to enable tabular deep learning prediction when $d \gg n$.

## 1 Introduction

Machine learning models have reached state-of-the-art performance in domains with abundant labeled data like computer vision (Wortsman et al., 2022; Deng et al., 2009) and natural language processing (Wang et al., 2019; Devlin et al., 2019; Ramesh et al., 2022). However, for tabular datasets in which the number $d$ of features vastly exceeds the number $n$ of samples, machine learning models struggle to achieve strong performance (Hastie et al., 2009; Liu et al., 2017). Unfortunately, many high impact domains like chemistry (Guyon et al., 2004), biology (Iorio et al., 2016; Yang et al., 2012; Garnett et al., 2012; Gao et al., 2015), and physics (Kasieczka et al., 2021) produce datasets with high-dimensional features but limited labeled samples due to the high time and labor costs associated with experiments. In chemistry, for example, mass spectrometry datasets can have tens of thousands of features but only tens or hundreds of samples (Guyon et al., 2004). For these and other tabular datasets with $d \gg n$, the performance of machine learning systems is currently limited.

To date, deep learning approaches for tabular data have focused on data regimes with far more samples than features ($n \gg d$) (Grinsztajn et al., 2022; Gorishniy et al., 2021; Shwartz-Ziv & Armon, 2022). In the low-data regime with far more features than samples ($d \gg n$), the dominant approaches are classical statistical methods (Hastie et al., 2009). These statistical methods reduce the dimensionality of the input space(Abdi & Williams, 2010; Liu et al., 2017; Van der Maaten & Hinton, 2008; Van Der Maaten et al., 2009), select features (Tibshirani, 1996; Climente-González et al., 2019; Freidling et al., 2021; Meier et al., 2008), impose regularization penalties on parameter magnitudes (Marquardt & Snee, 1975), or use ensembles of weak tree-based models (Friedman, 2001; Chen & Guestrin, 2016; Ke et al., 2017; Lou & Obukhov, 2017; Prokhorenkova et al., 2018).

Here, we present a novel problem setting and framework for tabular deep learning when $d \gg n$ (Figure 1). Our key insight is that even in tabular settings with limited labeled data, input features often represent real-world entities about which there is abundant prior information which can be structured as an auxiliary knowledge graph (KG). We propose a novel problem setting in which every input feature of a tabular dataset corresponds to a node in an auxiliary KG (Figure 1a). For example, consider a tabular medical dataset in which every row is a cancer patient, every column is a gene, every value is the amount of that gene in the patient's tumor, and the task is to predict the patient's survival. For this tabular dataset, there exists an auxiliary KG which consists of each gene's function, the relationships between genes, how a gene affects a part of human anatomy, and how human anatomy itself is structured. Note that the KG does *not* capture the relationships between *input data samples* but instead captures the relationships between *input features*.

Within our novel problem setting, we propose PLATO, a deep learning method for tabular data with $d \gg n$ and an auxiliary KG with input features as nodes (Figure 1(b)-(e)). PLATO uses a modified multilayer perceptron (MLP) to predict the output labels from the input samples and the auxiliary KG with two methodological components. First, the parameters in the first layer of the MLP are predicted from the auxiliary KG and the input sample rather than learned from just the tabular data. PLATO thereby integrates prior information about the input features from the auxiliary KG and drastically reduces the number of trainable parameters in the MLP. Second, the parameters in the first layer of the MLP are predicted differently for every sample by using the auxiliary KG and the sample values. PLATO thereby increases the representational capacity of the MLP and enables effective predictions.

We exhibit PLATO's performance on 6 datasets. We choose computational biology as it is a rich domain for $d \gg n$ in which we can construct a single knowledge graph to serve as a unified backbone for many distinct tabular datasets with distinct input features. We compare PLATO to 10 state-of-the-art baselines spanning dimensionality reduction, feature selection, classic statistical models, deep tabular learning methods, and parameter-prediction methods. Following a rigorous evaluation protocol from the tabular deep learning literature (Grinsztajn et al., 2022; Gorishniy et al., 2021), PLATO achieves or matches the prior state-of-the-art on all 6 datasets, achieving performance improvements of up to 10.19%. Ablation studies further demonstrate the necessity of each methodological component of PLATO. Ultimately, PLATO uses an auxiliary KG about input features to enable tabular deep learning prediction when $d \gg n$.

## 2 RELATED WORK

**Tabular deep learning methods.** In contrast to PLATO's setting, tabular deep learning methods have been developed for settings with far more samples than features (*i.e.* $n \gg d$). Recent tabular deep learning benchmarks ignore datasets with high numbers of features and low numbers of samples (Grinsztajn et al., 2022; Gorishniy et al., 2021; Shwartz-Ziv & Armon, 2022). In the $n \gg d$ setting, various categories of deep tabular models have been benchmarked. We select the state-of-the-art models to compare against PLATO. First, decision tree models like NODE (Popov et al., 2020) make decision trees differentiable to enable gradient-based optimization (Hazimeh et al., 2020; Kontschieder et al., 2015; Yang et al., 2018). Second, tabular transformer architectures use an attention mechanism to select and learn interactions among features. These include TabNet (Arik & Pfister, 2021), TabTransformer (Huang et al., 2020), FT-Transformer (Gorishniy et al., 2021), and others (Song et al., 2019; Somepalli et al., 2021; Kossen et al., 2021).

$d \gg n$ **methods.** For PLATO's setting in which $d \gg n$, various tabular machine learning approaches exist (Hastie et al., 2009). First, dimensionality reduction techniques like PCA (Abdi & Williams, 2010) aim to reduce the dimensionality of the input data while preserving as much of the the variance in the data as possible (Liu et al., 2017; Van der Maaten & Hinton, 2008; Van Der Maaten et al., 2009). Second, feature selection approaches select a parsimonious set of features, leading to a smaller feature space. Classical feature selection approaches include LASSO (Tibshirani, 1996) and its variants (Climente-González et al., 2019; Freidling et al., 2021; Meier et al., 2008). For feature selection with deep learning, Stochastic Gates (Yamada et al., 2020) are among the best performing of many variants (Balın et al., 2019; Lu et al., 2018). Finally, classical tree-based models like XGBoost learn ensembles of weak decision trees models to make an overall prediction (Friedman, 2001; Chen & Guestrin, 2016; Ke et al., 2017; Prokhorenkova et al., 2018).

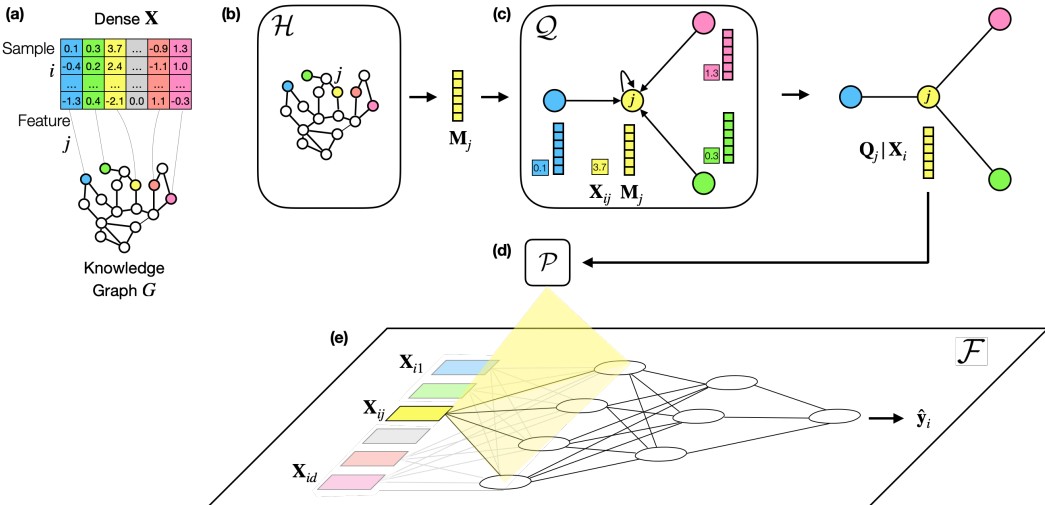

**Figure 1: PLATO is a machine learning model for tabular data with $d \gg n$ and an auxiliary knowledge graph with input features as nodes.** Machine learning models struggle to achieve strong performance on tabular datasets with far more $d$ features than $n$ samples (*i.e.* $d \gg n$). **(a)** The key insight of PLATO is that even in settings with limited labeled samples, input features often represent real-world entities about which there is abundant prior knowledge. We propose a new problem setting in which every feature in the input matrix corresponds to a node in an auxiliary knowledge graph (KG) $G$. **(b-e)** PLATO uses $G$ to predict the parameters in the first layer of a modified MLP $\mathcal{F}$. **(b)** First, PLATO pretrains an embedding $\mathbf{M}_j \in \mathbb{R}^c$ for every feature node in $G$ using $\mathcal{H}$, a self-supervised KG node embedding approach. **(c)** Second, PLATO updates each feature embedding to focus on the feature information that is most relevant to an input sample $\mathbf{X}_i$. PLATO uses a message passing network $\mathcal{Q}$ to produce $\mathbf{Q}_j \in \mathbb{R}^c$. $\mathcal{Q}$ uses an attention mechanism which considers the input sample $\mathbf{X}_i$. $\mathbf{Q}_j$ thus depends on the input sample (*i.e.* $\mathbf{Q}_j | \mathbf{X}_i$). **(d, e)** Finally, PLATO uses a small neural network $\mathcal{P}$ to predict the parameters in the first layer of a MLP $\mathcal{F}$ from $\mathbf{Q}_j$. The parameters in the first layer of $\mathcal{F}$ vary for every input sample $\mathbf{X}_i$.

**Knowledge graph approaches.** Existing knowledge graph approaches are designed for tasks directly on the graph such as link prediction (Wang et al., 2017; Trouillon et al., 2016; Wang et al., 2014; Yang et al., 2015; d'Amato et al., 2021). By contrast, PLATO does not make any predictions on the knowledge graph. Instead, PLATO makes predictions on a separate, tabular dataset by using the knowledge graph as prior information about the features and domain.

**Graph classification approaches.** In graph classification models, every input sample is a graph with node attributes, and a model must make a prediction for that graph. Graph classification models are not relevant for PLATO's problem setting. Graph classification models assume that different samples correspond to different graphs (Ying et al., 2021; Hu et al., 2020b;a). However, in PLATO every input sample corresponds to the exact same graph. Additional comments are in Appendix B.

**Parameter prediction.** Using one network to predict the parameters of another has been extensively studied (Denil et al., 2013; Schmidhuber, 1992; Bengio et al., 1991). For example, Ha et al. (2016) predicts the weights in all layers of a sequential model (*i.e.* RNN, LSTM) by using information about the structure of the weights. Another approach, Diet Networks (Romero et al., 2017) predicts parameters by hand-crafting prior information about the input features or using random projections. By contrast, PLATO predicts parameters in a network by leveraging prior information about the input features in an auxiliary KG. PLATO systematically constructs an embedding for each input feature which contains the prior information about the feature that is most relevant to a given sample.

## 3 PLATO

PLATO is a machine learning method for tabular datasets with $d \gg n$ and an auxiliary KG with input features as nodes (Section 3.1). The key insight of PLATO is that even in tabular datasets with limited labeled samples, input features often represent real world entities about which there is abundant prior information which can be structured as an auxiliary knowledge graph (KG) $G$ (Figure 1(a)).

PLATO uses a modified MLP to predict the output labels from the input samples and the auxiliary KG. PLATO's modified MLP has two key methodological components. First, the parameters in the first layer of the MLP are predicted from the auxiliary KG and the input sample rather than learned from just the tabular data (Figure 1(b-e)). PLATO thereby integrates prior information about the input features from the auxiliary KG and drastically reduces the number of trainable parameters in the MLP. Second, the parameters in the first layer of the MLP are predicted differently for every sample by using the auxiliary KG and the sample values. PLATO thereby increases the representational capacity of the MLP and enables effective predictions. The full PLATO Algorithm is given in Algorithm 1.

## 3.1 PROBLEM SETTING

Consider a tabular dataset $\mathbf{X} \in \mathbb{R}^{n \times d}$ with labels $\mathbf{y} \in \mathbb{R}^n$ and far more features than samples such that $d \gg n$. The goal is to train a machine learning model $\mathcal{F}$ to predict labels $\hat{\mathbf{y}}$ from the input $\mathbf{X}$. PLATO assumes the existence of an auxiliary knowledge graph $G = (V, E)$ with $|V|$ nodes and $|E|$ edges such that every input feature $j$ corresponds to a node in $G$. Formally, $\forall j \in \{1, \ldots, d\}$, $\exists v \in V$ s.t. $j \mapsto v$, as shown in Figure 1(a). $G$ also contains additional nodes which represent broader knowledge about the domain. The edges in $G$ are (head node, relation type, tail node) triplets.

## 3.2 THE PLATO MLP $\mathcal{F}$

Consider a standard MLP $\hat{\mathbf{y}} = \mathcal{T}(\mathbf{X}; \mathbf{\Theta})$ with $L$ layers, $h$ hidden units in the first layer, and trainable parameters $\mathbf{\Theta} = \{\mathbf{\Theta}^{[1]}, \mathbf{\Theta}^{[2]}, \ldots, \mathbf{\Theta}^{[L]}\}$. The PLATO MLP $\mathcal{F}$ differs from $\mathcal{T}$ in two key ways.

**First, the parameters in the first layer of PLATO's MLP $\mathcal{F}$ are predicted from prior information rather than learned only from the tabular data.** We observe that every parameter in the first layer of $\mathcal{T}$ is associated with an input feature $j$. In particular, $\mathbf{\Theta}^{[1]} \in \mathbb{R}^{d \times h}$ such that $\mathbf{\Theta}_j^{[1]} \in \mathbb{R}^h$ is a vector of parameters connecting input feature $j$ to every hidden unit in the first layer of the MLP (Figure 1e). Typically, $\mathcal{T}$ learns the parameters $\mathbf{\Theta}_j^{[1]}$ and $\mathbf{\Theta}_k^{[1]}$ associated with two features $j$ and $k$ independently by gradient backpropagation. In PLATO, we propose that if two input features $j$ and $k$ represent real-world entities that are related, then their corresponding parameters $\mathbf{\Theta}_j^{[1]}$ and $\mathbf{\Theta}_k^{[1]}$ should be related too. To capture the intuition that related input features $j$ and $k$ should have related parameters, PLATO's MLP $\mathcal{F}$ predicts $\hat{\mathbf{\Theta}}_j^{[1]}$ and $\hat{\mathbf{\Theta}}_k^{[1]}$ from prior information known about $j$ and $k$. If input features $j$ and $k$ are related, then the parameter prediction module produces related $\hat{\mathbf{\Theta}}_j^{[1]}$ and $\hat{\mathbf{\Theta}}_k^{[1]}$. Parameter prediction details are in Section 3.3. For now, note that the parameters in the first layer of PLATO's MLP $\mathcal{F}$ are predicted such that $\hat{\mathbf{\Theta}}^{[1]} \in \mathbb{R}^{d \times h}$. The parameters $\mathbf{\Theta}^{[2]}, \ldots, \mathbf{\Theta}^{[L]}$ in the remaining layers of PLATO's MLP $\mathcal{F}$ are learned normally.

**Second, the parameters in the first layer of PLATO's MLP are allowed to vary for every input sample.** In the standard MLP $\mathcal{T}$, all parameters $\mathbf{\Theta}^{[1]}, \ldots, \mathbf{\Theta}^{[L]}$ are the same for every input sample $\mathbf{X}_i$. In the first layer of PLATO's MLP $\mathcal{F}$, however, the parameters $\hat{\mathbf{\Theta}}^{[1]}$ are being predicted from prior information about the input features. We observe that for each input sample $\mathbf{X}_i$, the most relevant prior information about each input feature $j$ might differ. Therefore for each sample $\mathbf{X}_i$, PLATO uses different prior information about each input feature $j$ to predict the parameters $\hat{\mathbf{\Theta}}_j^{[1]}$. As a result, the parameters $\hat{\mathbf{\Theta}}^{[1]}$ in the first layer of $\mathcal{F}$ vary with each input sample $\mathbf{X}_i$, increasing the representational capacity of $\mathcal{F}$. How PLATO uses different prior information for parameter prediction is left to Section 3.3. For now, note that PLATO predicts $\hat{\mathbf{\Theta}}_j^{[1]}$ from prior information about feature $j$. The prior information about feature $j$ that is used depends on the input sample $\mathbf{X}_i$. As a result, $\hat{\mathbf{\Theta}}^{[1]}$ is conditional on $\mathbf{X}_i$ according to $\hat{\mathbf{\Theta}}^{[1]}|\mathbf{X}_i$.

**Formal Notation.** Overall, PLATO's MLP $\mathcal{F}$ takes the form

$$\hat{\mathbf{y}}_i = \mathcal{F}(\mathbf{X}_i; \hat{\mathbf{\Theta}}|\mathbf{X}_i). \tag{1}$$

$\mathcal{F}$ has parameters $\hat{\mathbf{\Theta}} = \{\hat{\mathbf{\Theta}}^{[1]}|\mathbf{X}_i\} \cup \{\mathbf{\Theta}^{[2]}, \ldots, \mathbf{\Theta}^{[L]}\}$ where $L$ is the number of layers in $\mathcal{F}$. The parameters $\hat{\mathbf{\Theta}}^{[1]}|\mathbf{X}_i$ in the first layer of $\mathcal{F}$ are predicted from the input sample $\mathbf{X}_i$ via message-passing on the KG according to Section 3.3. For every sample $i$, a new $\hat{\mathbf{\Theta}}^{[1]}$ is predicted such that $\hat{\mathbf{\Theta}}^{[1]}$

is conditional on $\mathbf{X}_i$ at both training and inference time. The dimensionality of $\hat{\boldsymbol{\Theta}}^{[1]} \in \mathbb{R}^{d \times h}$ is the same as in a normal MLP where $h$ is the number of hidden units in the first layer of $\mathcal{F}$. The parameters in the remaining layers of PLATO's MLP $\boldsymbol{\Theta}^{[2]}, \ldots, \boldsymbol{\Theta}^{[L]}$ are the same as in a standard MLP: they are learned normally, are the same for every sample at both training and inference time, and are thus not conditional on $\mathbf{X}_i$.

## 3.3 PREDICTING THE PARAMETERS IN THE FIRST LAYER OF PLATO'S MLP $\mathcal{F}$

PLATO uses prior information about the input features to predict the parameters in the first layer of PLATO's MLP $\mathcal{F}$. PLATO predicts these parameters in three steps. First, PLATO uses self-supervision on the auxiliary KG to pretrain an embedding for every input feature (Section 3.3.1, Figure 1(b)). Second, since different input samples might rely on different prior information about each input feature, PLATO updates each feature embedding to contain the most relevant prior information about the input feature for the given input sample (Section 3.3.2, Figure 1(c)). Finally, PLATO predicts the parameters in the first layer of $\mathcal{F}$ from the updated feature embeddings with a small neural network that is shared across input features (Section 3.3.3, Figure 1(d)(e)).

### 3.3.1 PRETRAINING FEATURE EMBEDDINGS WITH SELF-SUPERVISION

First, PLATO learns general prior information about each input feature $j$ from the auxiliary KG $G$ (Figure 1b). PLATO represents the general prior information about each input feature $j$ as a low-dimensional embedding $\mathbf{M}_j \in \mathbb{R}^c$. Since every input feature $j$ corresponds to a specific node in $G$, PLATO can learn $\mathbf{M}_j$ by learning an embedding for the corresponding feature node in $G$. Any self-supervised node embedding method on $G$ can be used within PLATO's framework.

**Formal notation.** Formally, PLATO uses self-supervision on $G$ to pretrain an embedding for every input feature according to

$$\mathbf{M} = \mathcal{H}(G). \tag{2}$$

$\mathbf{M} \in \mathbb{R}^{d \times c}$ is the matrix of all feature embeddings. $\mathcal{H}$ is a self-supervised node embedding method.

We refer to Eq. (2) as pretraining since only the auxiliary KG $G$ is used but the tabular data $\mathbf{X}, \mathbf{y}$ is ignored. After pretraining, the feature embeddings $\mathbf{M}$ are fixed.

**Implementation.** For $\mathcal{H}$, we choose ComplEx as it is prominent and highly scalable KG node embedding method (Trouillon et al., 2016). ComplEx uses a self-supervised objective which learn an embedding for every node in $G$ by classifying whether a proposed edge exists in $G$. ComplEx's proposed edges include both feature nodes and other nodes in $G$, thereby integrating prior information about the input features and the broader domain.

### 3.3.2 UPDATING FEATURE EMBEDDINGS TO CONTAIN THE MOST RELEVANT INFORMATION FOR AN INPUT SAMPLE

Since different input samples might rely on different prior information about each input feature, PLATO next updates each feature embedding $\mathbf{M}_j \in \mathbb{R}^{d \times c}$ to $\mathbf{Q}_j \in \mathbb{R}^{d \times c}$, a feature embedding which contain the most relevant prior information about feature $j$ for a given input sample $\mathbf{X}_i$ (Figure 1(c)). PLATO uses a message-passing network $\mathcal{Q}$ on the KG to update the feature embeddings in a way that minimizes the number of additional trainable parameters.

$$\mathbf{Q} = \mathcal{Q}(\mathbf{X}_i, \mathbf{M}, G; \boldsymbol{\Pi}). \tag{3}$$

The message-passing network in $\mathcal{Q}$ uses an attention mechanism which considers the sample values $\mathbf{X}_i$ to update the feature embeddings. The attention mechanism has a small number of trainable parameters $\boldsymbol{\Pi}$.

**The message passing network $\mathcal{Q}$.** Let $\mathbf{Q}_j^{[r]}$ be the embedding of input feature $j$ after round $r \in \{1, ..., R\}$ of message passing. For every input feature $j$, $\mathcal{Q}$ first initializes the updated feature embedding to the pretrained feature embedding.

$$\mathbf{Q}_j^{[0]} = \mathbf{M}_j. \tag{3a}$$

$\mathcal{Q}$ then conducts $R$ rounds of message passing. In each round of message passing, the feature embedding $\mathbf{Q}_j^{[r]}$ is updated from the feature embedding of each neighbor $k$ in the prior round $\mathbf{Q}_k^{[r-1]}$ and its own feature embedding in the prior round $\mathbf{Q}_j^{[r-1]}$. The "message" being passed is the embedding of each feature from the prior round.

$$\mathbf{Q}_j^{[r]} = \sigma\left[ \overbrace{\beta(\sum_{k \in N_j} \mathbf{A}_{ijk}\mathbf{Q}_k^{[r-1]})}^{\text{Weighted messages from neighbors}} + \underbrace{(1-\beta)\mathbf{Q}_j^{[r-1]}}_{\text{Weighted message from self}} \right]. \tag{3b}$$

$$\mathbf{Q}_j = \sigma\left[ \beta(\sum_{k \in N_j} \mathbf{A}_{ijk}\mathbf{M}_k) + (1-\beta)\mathbf{M}_j \right]. \tag{3c}$$

$\sigma$ is an optional nonlinearity. $N_j$ are the neighbors of feature node $j$ in $G$.

During message-passing, $\mathcal{Q}$ uses two scalar values $\beta \in \mathbb{R}$ and $\mathbf{A}_{ijk} \in \mathbb{R}$ to control the weights of messages. First, $\mathcal{Q}$ uses hyperparameter $\beta$ to control the weight of the messages aggregated from all neighbors vs. the message from the feature node $j$ itself. Second, $\mathcal{Q}$ calculates an attention score $\mathbf{A}_{ijk}$ to control the weight of the specific message between feature $j$ and neighbor $k$. The attention score is different for every sample $i$ and is calculated by a shallow neural network $\mathcal{A}$ with a small number of trainable parameters $\mathbf{\Pi}$. The attention score $A_{ijk}$ thus enables $\mathcal{Q}$ to update the information in the feature embedding in a way that is most relevant for the input sample $i$. Formally:

$$\mathbf{A}_{ijk} = \frac{\exp\left(\mathcal{A}(\mathbf{X}_{ij}, \mathbf{X}_{ik}; \mathbf{\Pi})\right)}{\sum_{t \in N_j} \exp\left(\mathcal{A}(\mathbf{X}_{ij}, \mathbf{X}_{it}; \mathbf{\Pi})\right)}. \tag{3d}$$

The number of trainable parameters in $\mathbf{\Pi}$ is small since the input of $\mathcal{A}$ is $\mathbb{R}^2$ and the output of $\mathcal{A}$ is a scalar $\mathbb{R}$. $\mathcal{A}$ and its parameters $\mathbf{\Pi}$ are shared for all samples and features.

Finally, the updated feature embeddings $\mathbf{Q}_j$ are set after $R$ rounds of message-passing.

$$\mathbf{Q}_j = \mathbf{Q}_j^{[R]}. \tag{3e}$$

### 3.3.3 PREDICTING THE FIRST LAYER OF PARAMETERS IN $\mathcal{F}$ FROM THE UPDATED FEATURE EMBEDDINGS

Finally, PLATO predicts the parameters in the first layer of $\mathcal{F}$ from each updated feature embedding (Figure 1(d)(e)). Every parameter in the first layer of $\mathcal{F}$ is associated with a feature $j$. PLATO thus predicts $\hat{\mathbf{\Theta}}_j$, the parameters associated with the feature $j$, from $\mathbf{Q}_j$, the prior information about $j$.

**Formal notation.** PLATO predicts the parameters associated with every input feature $j$ in the first layer of $\mathcal{F}$ according to

$$\hat{\mathbf{\Theta}}_j^{[1]} = \mathcal{P}(\mathbf{Q}_j | \mathbf{X}_i; \mathbf{\Phi}). \tag{4}$$

$\mathcal{P}$ is a shallow neural network parameterized by $\mathbf{\Phi}$. $\mathbf{Q}_j$ is the updated feature embedding of $j$ which is conditional on the specific input sample $\mathbf{X}_i$. $\mathbf{\Phi}$ are the parameters of $\mathcal{P}$. $\mathcal{P}$ and its parameters $\mathbf{\Phi}$ are shared for every feature $j \in \{1, \ldots, d\}$.

**PLATO drastically reduces the number of trainable parameters compared to a standard MLP.** The sharing of $\mathcal{P}$ and $\mathbf{\Phi}$ across all input features enables a drastic reduction in the number of trainable parameters compared to a standard MLP. For a high-dimensional tabular dataset (*i.e.* $d \gg n$), a standard MLP $\mathcal{T}$ with $h$ hidden units has a large number of trainable parameters in the first layer since $\mathbf{\Theta}^{[1]} \in \mathbb{R}^{d \times h}$. A standard MLP $\mathcal{T}$ must learn all $dh$ of these trainable parameters independently. By contrast, $\mathcal{P}$ uses a shared set of trainable parameters $\mathbf{\Phi}$ to predict $\hat{\mathbf{\Theta}}_j$ from $\mathbf{Q}_j$ for every $j \in \{1, \ldots, d\}$. The number of trainable parameters in $\Phi$ is small compared to $dh$ since $\mathcal{P}$ need only transform every $\mathbf{Q}_j \in \mathbb{R}^c$ to $\hat{\mathbf{\Theta}}^{[1]} \in \mathbb{R}^h$. Thus, $|\mathbf{\Phi}| = ch$ (assuming that $\mathcal{P}$ is a single layer neural network). $c$, the dimensionality of the feature embedding, is much less than $d$ the number of input features. As a result, $|\mathbf{\Phi}| = ch \ll dh$ and PLATO drastically reduces the number of trainable parameters in the first layer of a MLP.

---

**Algorithm 1:** The PLATO Algorithm.

---

**Input:** a data sample $\mathbf{X}_i \in \mathbb{R}^d$, a knowledge graph (KG) $G = (V, E)$
**Output:** predicted label $\hat{\mathbf{y}}_i \in \mathbb{R}$

1 Pretrain KG embedding for every feature: $\mathbf{M} = \mathcal{H}(G)$

2 Initialize feature embedding for feature $j$: $\mathbf{Q}_j^{[0]} = \mathbf{M}_j$

3 Compute sample $i$-specific attention weight: $\mathbf{A}_{ijk} = \frac{\exp\left(\mathcal{A}(\mathbf{X}_{ij}, \mathbf{X}_{ik}; \mathbf{\Pi})\right)}{\sum_{t \in N_j} \exp\left(\mathcal{A}(\mathbf{X}_{ij}, \mathbf{X}_{it}; \mathbf{\Pi})\right)}, \forall$ features $j, k$,
   where $\mathcal{A}$ is a NN parameterized by $\mathbf{\Pi}$

4 **for** $r = 1;\ r \leq R$ **do**

5   Update feature embedding with message passing neural network at layer $r$:
   $$\mathbf{Q}_j^{[r]} = \sigma\left[\beta(\sum_{k \in N_j} \mathbf{A}_{ijk}\mathbf{Q}_k^{[r-1]}) + (1 - \beta)\mathbf{Q}_j^{[r-1]}\right]$$

6 **end**

7 Obtain feature $j$ embedding from GNN last layer $R$: $\mathbf{Q}_j = \mathbf{Q}_j^{[R]}$

8 Predict the parameter of first layer of a MLP from the feature embedding: $\hat{\mathbf{\Theta}}^{[1]} = \mathcal{P}(\mathbf{Q}|\mathbf{X}_i; \mathbf{\Phi})$,
   where $\mathcal{P}$ is a NN parameterized by $\mathbf{\Phi}$

9 Concatenate the first layer predicted parameters with the parameters from the rest of layers:
   $\hat{\mathbf{\Theta}} = \{\hat{\mathbf{\Theta}}^{[1]}\} \cup \{\mathbf{\Theta}^{[2]}, \ldots, \mathbf{\Theta}^{[L]}\}$

10 Predict label: $\hat{\mathbf{y}}_i = \mathcal{F}(\mathbf{X}; \hat{\mathbf{\Theta}}|\mathbf{X}_i)$, where $\mathcal{F}$ is an MLP parameterized by $\hat{\mathbf{\Theta}}$

---

**Table 1: PLATO outperforms statistical and deep baselines when $d \gg n$.** For every dataset, the best overall model is in **bold** and the second best model is underlined.

| Dataset | | MNSCLC | CM | PDAC | BRCA | CRC | CH |
|---|---|---|---|---|---|---|---|
| D | | 15,390 | 13,183 | 12,932 | 12,693 | 18,206 | 19,902 |
| N | | 295 | 286 | 321 | 476 | 562 | 924 |
| D/N | | 52.2 | 46.1 | 40.3 | 28.2 | 22.6 | 19.7 |
| Classic Stat ML | Ridge | $0.153_{\pm 0.000}$ | $0.390_{\pm 0.000}$ | $0.344_{\pm 0.000}$ | $\underline{0.538}_{\pm 0.000}$ | $0.376_{\pm 0.000}$ | $0.546_{\pm 0.000}$ |
| Dim. Reduct. | PCA | $0.156_{\pm 0.113}$ | $0.070_{\pm 0.000}$ | $0.232_{\pm 0.121}$ | $0.452_{\pm 0.000}$ | $0.193_{\pm 0.163}$ | $0.237_{\pm 0.232}$ |
| Feat. Select. | LASSO | $0.168_{\pm 0.000}$ | $\underline{0.431}_{\pm 0.000}$ | $0.346_{\pm 0.000}$ | $0.470_{\pm 0.000}$ | $\underline{0.400}_{\pm 0.000}$ | $0.547_{\pm 0.000}$ |
| | STG | $0.132_{\pm 0.130}$ | $0.366_{\pm 0.043}$ | $0.258_{\pm 0.055}$ | $0.485_{\pm 0.037}$ | $0.301_{\pm 0.010}$ | $0.262_{\pm 0.076}$ |
| Decision Tree | XGBoost | $-0.02_{\pm 0.000}$ | $0.225_{\pm 0.000}$ | $\underline{0.363}_{\pm 0.000}$ | $0.347_{\pm 0.000}$ | $0.354_{\pm 0.000}$ | $\underline{0.728}_{\pm 0.000}$ |
| Param. Pred. | Diet | $-0.04_{\pm 0.205}$ | $0.054_{\pm 0.149}$ | $0.309_{\pm 0.096}$ | $0.213_{\pm 0.036}$ | $0.087_{\pm 0.112}$ | $0.148_{\pm 0.008}$ |
| Tabular DL | MLP | $0.128_{\pm 0.126}$ | $0.322_{\pm 0.043}$ | $0.289_{\pm 0.047}$ | $0.240_{\pm 0.067}$ | $0.355_{\pm 0.022}$ | $0.044_{\pm 0.039}$ |
| | NODE | $0.003_{\pm 0.000}$ | $0.150_{\pm 0.000}$ | $0.190_{\pm 0.000}$ | $0.512_{\pm 0.000}$ | $0.344_{\pm 0.000}$ | $0.181_{\pm 0.000}$ |
| | TabTransformer | $\underline{0.265}_{\pm 0.000}$ | $0.072_{\pm 0.000}$ | $0.029_{\pm 0.000}$ | $0.202_{\pm 0.000}$ | $0.238_{\pm 0.000}$ | $0.020_{\pm 0.000}$ |
| | TabNet | $0.085_{\pm 0.028}$ | $0.010_{\pm 0.068}$ | $0.088_{\pm 0.037}$ | $0.055_{\pm 0.037}$ | $0.018_{\pm 0.016}$ | $0.039_{\pm 0.026}$ |
| Ours | PLATO | $\mathbf{0.272}_{\pm 0.130}$ | $\mathbf{0.435}_{\pm 0.022}$ | $\mathbf{0.400}_{\pm 0.021}$ | $\mathbf{0.583}_{\pm 0.019}$ | $\mathbf{0.401}_{\pm 0.019}$ | $\mathbf{0.770}_{\pm 0.003}$ |

## 4 EXPERIMENTS

We evaluate PLATO against 10 statistical and deep baselines on 6 tabular datasets with $d \gg n$.

**Datasets.** We use 6 tabular $d \gg n$ datasets in biomedicine compiled from prior studies (Gao et al., 2015; Garnett et al., 2012; Iorio et al., 2016; Yang et al., 2012). We focus on biomedicine because it is a rich domain for $d \gg n$ in which a single KG can be used as a unified knowledge backbone across many datasets. Additional data descriptions are in Appendix E.

**Auxiliary Knowledge Graph.** As a unified knowledge backbone for the datasets, we compile a general biomedical knowledge graph from prior studies (et al., 2020; 2016; Kuhn et al., 2015; Ruiz et al., 2021; Szklarczyk et al., 2020; Wishart et al., 2017a;b). Our knowledge graph contains 108,447 nodes, 3,066,156 edges, and 99 relation types. All datasets include features which map to a subset of nodes in the knowledge graph. The remaining nodes serve as broader domain knowledge. The same KG is used across all datasets even though all datasets have distinct feature sets with distinct cardinalities. PLATO thus allows a single KG to serve as a unified knowledge backbone for different datasets in a domain. Additional KG details are in Appendix F.

Table 2: PLATO's performance depends on updating feature embeddings to contain information that is specific to a given sample.

| Parameter Pred. $\mathcal{P}$ Input | Feature Information | Sample Specific | PearsonR |
|---|---|---|---|
| Updated feat. embed. $\mathbf{Q}$ | ✔ | ✔ | $0.583_{\pm 0.019}$ |
| General feat. embed $\mathbf{M}$ | ✔ | ✗ | $0.522_{\pm 0.030}$ |
| None | ✗ | ✗ | $0.240_{\pm 0.067}$ |

Table 3: PLATO's performance depends on both feature nodes in $G$ and other nodes which represent broader domain knowledge.

| Auxiliary KG | Feature-only Knowledge | Broader Knowledge | PearsonR |
|---|---|---|---|
| Full KG | ✔ | ✔ | $0.583_{\pm 0.019}$ |
| Feature-only KG | ✔ | ✗ | $0.539_{\pm 0.038}$ |
| No KG | ✗ | ✗ | $0.240_{\pm 0.067}$ |

Table 4: PLATO's performance is competitive with baselines when $d \sim n$. For every dataset, the best overall model is in **bold** and the second best model is underlined.

| Dataset | | ME | BC | SCLC | NSCLC |
|---|---|---|---|---|---|
| D | | 19,902 | 10,101 | 10,712 | 16,730 |
| N | | 19,902 | 18,261 | 18,437 | 18,308 |
| D/N | | 2.0 | 1.8 | 1.7 | 1.1 |
| Classic Stat ML | Ridge | $0.566_{\pm 0.008}$ | $0.483_{\pm 0.008}$ | $0.604_{\pm 0.057}$ | $0.679_{\pm 0.008}$ |
| Dim. Reduct. | PCA | $0.239_{\pm 0.310}$ | $0.233_{\pm 0.294}$ | $0.284_{\pm 0.274}$ | $0.645_{\pm 0.000}$ |
| Feat. Select. | LASSO | $0.667_{\pm 0.000}$ | $0.633_{\pm 0.000}$ | $0.669_{\pm 0.000}$ | $0.637_{\pm 0.000}$ |
| | STG | $0.676_{\pm 0.000}$ | $0.643_{\pm 0.000}$ | $0.668_{\pm 0.000}$ | $0.646_{\pm 0.000}$ |
| Decision Tree | XGBoost | $\mathbf{0.875}_{\pm \mathbf{0.000}}$ | $\underline{0.826}_{\pm 0.000}$ | $\underline{0.878}_{\pm 0.000}$ | $\mathbf{0.843}_{\pm \mathbf{0.000}}$ |
| Param. Pred. | Diet | $0.105_{\pm 0.000}$ | $0.037_{\pm 0.000}$ | $-0.05_{\pm 0.000}$ | $0.002_{\pm 0.000}$ |
| Tabular DL | MLP | $0.487_{\pm 0.131}$ | $0.508_{\pm 0.061}$ | $0.537_{\pm 0.061}$ | $0.573_{\pm 0.005}$ |
| | NODE | $0.870_{\pm 0.000}$ | $0.420_{\pm 0.169}$ | $0.801_{\pm 0.102}$ | $0.487_{\pm 0.197}$ |
| | TabTransformer | $0.305_{\pm 0.028}$ | $0.010_{\pm 0.000}$ | $0.288_{\pm 0.203}$ | $0.503_{\pm 0.187}$ |
| | TabNet | $0.667_{\pm 0.002}$ | $0.624_{\pm 0.001}$ | $0.657_{\pm 0.004}$ | $0.647_{\pm 0.000}$ |
| Ours | PLATO | $\mathbf{0.875}_{\pm \mathbf{0.004}}$ | $\mathbf{0.844}_{\pm \mathbf{0.003}}$ | $\mathbf{0.883}_{\pm \mathbf{0.002}}$ | $\underline{0.839}_{\pm 0.000}$ |

**Baselines.** We compare PLATO to 10 state-of-the art statistical and deep baselines. We consider classic regularization with Ridge Regression (Marquardt & Snee, 1975), dimensionality reduction with PCA (Abdi & Williams, 2010), feature selection with LASSO (Tibshirani, 1996) deep feature selection with Stochastic Gates (Yamada et al., 2020), and gradient boosted decision trees with XGBoost (Chen & Guestrin, 2016). We also consider deep tabular learning methods including a standard MLP, self-attention-based tabular methods with TabTransformer (Huang et al., 2020) and TabNet (Arik & Pfister, 2021), differentiable decision trees with NODE (Popov et al., 2020), and parameter-prediction with Diet Networks (Romero et al., 2017). We also attempted FT-Transformer (Gorishniy et al., 2021), but it experienced out of memory issues on all datasets due to the large number of features.

**Fair Comparison of PLATO with Baselines.** To ensure a fair comparison with baselines, we follow evaluation protocols in tabular benchmarks (Grinsztajn et al., 2022; Gorishniy et al., 2021). We conduct a random search with 500 configurations of every model (including PLATO) on every dataset across a broad range of hyperparameters (Appendix A). We split data with a 60/20/20 training, validation, test split. All results are computed across 3 data splits and 3 runs of each model in each data split. We report the mean and standard deviation of the Pearson correlation (PearsonR) between $\mathbf{y}$ and $\hat{\mathbf{y}}$ across runs and splits on the test set. Each model is run on a GeForce RTX 2080 TI GPU.

## 4.1 RESULTS

**PLATO outperforms statistical and deep baselines when $d \gg n$.** PLATO outperforms all baselines across all 6 datasets with $d \gg n$ (Table 1). PLATO achieves the largest relative improvement on the PDAC dataset, improving by 10.19% vs. XGBoost, the best baseline for PDAC (0.400 vs. 0.363). While PLATO achieves the strongest performance across all 6 datasets, the best performing baseline varies with different datasets. Ridge Regression is the strongest baseline for BRCA, LASSO for CM and CRC, XGBoost for PDAC and CH, and TabTransformer for MNSCLC. The remaining baselines (PCA, STG, Diet Networks, MLP, NODE, and TabNet) are not the strongest baseline for any dataset. We also find that the performance of a specific baseline depends largely on the specific dataset. TabTransformer, for example, is the best baseline for the MNSCLC dataset but the worst baseline for the CH dataset. The rank order of all models on all datasets is Appendix C.

**PLATO's performance depends on updating feature embeddings to contain information relevant to a sample.** PLATO predicts the parameters $\hat{\Theta}^{[1]}$ in the first layer of a modified MLP $\mathcal{F}$ by using feature embeddings which contain prior information about the input features. PLATO first pretrains general feature embeddings $\mathbf{M} \in \mathbb{R}^{d \times c}$. PLATO then updates the feature embeddings to $\mathbf{Q} \in \mathbb{R}^{d \times c}$ which contains information about the input features that is most relevant to a given sample $\mathbf{X}_i$. We test whether updating the feature embeddings based on a given $\mathbf{X}_i$ is necessary by evaluating PLATO on the BRCA dataset in three configurations (Table 2). The default configuration uses the updated feature embeddings $\mathbf{Q}$ to predict $\hat{\Theta}^{[1]}$ according to $\hat{\Theta}^{[1]} = \mathcal{P}(\mathbf{Q}|\mathbf{X}_i)$. The second configuration uses the general feature embeddings $\mathbf{M}$ instead of $\mathbf{Q}$ to predict $\hat{\Theta}^{[1]}$ according to $\hat{\Theta}^{[1]} = \mathcal{P}(\mathbf{M})$. The third configuration does not use feature embeddings and thus reduces to a standard MLP.

We compare PLATO's performance when it uses feature embeddings which contain the relevant information for a given sample $\mathbf{X}_i$ (*i.e.* $\hat{\Theta}^{[1]} = \mathcal{P}(\mathbf{Q}|\mathbf{X}_i)$) vs. the pretrained feature embeddings $\mathbf{M}$ which contain general information about the input features that is not specific to a given sample (*i.e.* $\hat{\Theta}^{[1]} = \mathcal{P}(\mathbf{M})$). Using general feature embeddings $\mathbf{M}$ improves over not using feature embeddings at all (0.522 vs. 0.240). Using feature embeddings $\mathbf{Q}$ that are specific to a given input sample further improves performance (0.583 vs. 0.522). Therefore, updating the feature embeddings to $\mathbf{Q}$ such that they contain the information specific to a given sample is critical to PLATO's performance.

**PLATO's performance depends on both feature nodes and broader knowledge nodes in the auxiliary KG.** PLATO relies on an auxiliary KG $G$ which contains information about input features and information about the broader domain. Information about input features is represented as feature nodes while information about the broader domain is represented as other nodes in $G$ (Methods 3.1). To test the relative importance of the feature information in $G$ vs. the broader domain information, we measured the performance of PLATO on the BRCA dataset in two KG configurations: PLATO with the full KG (*i.e.* both the feature nodes and the broader domain nodes) and PLATO with a "feature-only KG" (*i.e.* an induced subgraph on only the feature nodes) (Table 3). We also compare to a "No KG" configuration in which PLATO does not have access to the KG. Without auxiliary information about the input features or the broader domain, PLATO is ablated to become a standard MLP.

We find that both the feature nodes and the broader knowledge nodes are important for PLATO's performance. Using the "feature-only KG" configuration of PLATO improves performance vs the "no KG" configuration of PLATO (0.539 vs 0.240). Using the "full KG" configuration further improves performance vs the "feature-only KG" configuration (0.583 vs 0.539). PLATO's performance thus relies on both the feature information and the broader domain information in the KG.

**For datasets with $d \sim n$, PLATO is competitive with baselines.** Finally, we test PLATO's performance for datasets with $d \sim n$. We test 4 datasets with $d \sim n$ ranging from $\frac{d}{n} = 1.1$ to $\frac{d}{n} = 2.0$ (Table 4). We find that on 4 datasets with $d \sim n$, PLATO is competitive with the best performing baseline, XGBoost, but does not improve performance substantially. PLATO's stronger performance for datasets with $d \gg n$ than for datasets with $d \sim n$ is justified. PLATO's key idea is to include auxiliary information about the input features. Auxiliary information is likely to help performance the most in settings with the least labeled data (*i.e.* $d \gg n$). When $d \sim n$, auxiliary information is less helpful since the tabular dataset may already have enough information to train a strong predictive model. We further find that XGBoost is consistently the strongest baseline for datasets with $d \sim n$, in contrast to the varied performance of XGBoost on the datasets with $d \gg n$ (Table 1).

## 5  DISCUSSION

PLATO is a machine learning model for tabular data with $d \gg n$ and an auxiliary KG with input features as nodes. Across 6 datasets and 10 baselines, PLATO achieves state-of-the-art performance, including relative performance improvements of up to $10.19\%$. Ablation studies also confirm the importance of each component of PLATO. PLATO has several limitations. First, PLATO matches but does not substantially improve the performance of baselines for high-dimensional datasets with more samples (*i.e.* $d \sim n$). Second, PLATO relies on the coverage of prior information. Datasets with input features that have little prior information in the KG are less likely to benefit from PLATO. Overall, PLATO uses an auxiliary KG about input features to enable tabular deep learning when $d \gg n$.

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

# Appendix

## A  EVALUATION PROTOCOL AND HYPERPARAMETER RANGES

To ensure a fair comparison with baselines, we follow evaluation protocols outlined in tabular benchmarks (Grinsztajn et al., 2022; Gorishniy et al., 2021). We conduct a random search with 500 configurations of every model (including PLATO) on every dataset across a broad range of hyperparameters. We base the hyperparameter ranges on the ranges used in prior tabular learning benchmarks (Grinsztajn et al., 2022; Gorishniy et al., 2021) and the ranges mentioned in the original papers of the methods. Hyperparameter ranges for PLATO are given in Supplementary Table 1. Hyperparameter ranges for baseline methods are given in Supplementary Table 2.

| Module in PLATO | Hyperparameter | Range |
|---|---|---|
| General | Learning rate
Batch size
L2 | LogUniform(1e-4, 5e-3)
[16, 32, 64]
0, LogUniform(1e-5, 1e-2) |
| KG $\mathcal{H}$ | Embedding dimension $c$
Embedding model | 200
ComplEx |
| Message Passing (MP) $\mathcal{Q}$ | # MP layers $R$
Beta
Hidden dimension in $\mathcal{A}$ | 2
LogUniform(1e-4, 1e-1)
UniformInt(16, 512) |
| Param Prediction $\mathcal{P}$ | # Layers $R$
Hidden dimension in $h$ | UniformInt(2, 6)
UniformInt(16, 512) |

**Supplementary Table 1: Hyperparameter ranges used for PLATO.**

| Model | Hyperparameter | Range |
|---|---|---|
| LASSO | L1 | LogUniform(1E-4, 10) |
| Ridge | L2 | LogUniform(1E-4, 10) |
| XGBoost | n-estimators | UniformInt(1,2000) |
| | Max depth | UniformInt(3, 10) |
| | Min weight | LogUniform(1E-8,1E5) |
| | Subsample | Uniform(0.5, 1) |
| | Learning rate | LogUniform(1E-5,1) |
| | Col sample by level | Uniform(0.5, 1) |
| | Col sample by tree | Uniform(0.5, 1) |
| | Gamma | 0, LogUniform(1E-8, 1E2) |
| | Lambda | 0, LogUniform(1E-8, 1E2) |
| | Alpha | 0, LogUniform(1E-8, 1E2) |
| | Booster | "gbtree" |
| | Early-stopping-rounds | 50 |
| | Iterations | 100 |
| PCA | Number of PCA Components | UniformInt(2,1000) |
| STG | Hidden dimension | UniformInt(10, 500) |
| | Number of layers | UniformInt(1, 5) |
| | Activation | [Tanh, Relu, Sigmoid] |
| | Learning rate | LogUniform(1e-4, 1e-1) |
| | Sigma | Uniform(0.001, 2) |
| | Lambda | LogUniform(1e-3, 10) |
| MLP | Number of layers | UniformInt(1, 8) |
| | Hidden dimension | UniformInt(1, 512) |
| | Dropout | 0, Uniform([0,0.5]) |
| | Learning rate | LogUniform(1e-5, 1e-2) |
| | L2 | 0, LogUniform(1e-6, 1e-3) |
| TabNet | Decision Steps | UniformInt(3, 10) |
| | Layer size | 2, 4, 8, 16, 32, 64 |
| | Relaxation factor | Uniform[1, 2] |
| | Sparsity loss weight | LogUniform[1e-6, 1e-1] |
| | Decay rate | Uniform[0.4, 0.95] |
| | Decay steps | 100, 500, 2000 |
| | Learning rate | Uniform(1e-3, 1e-2) |
| | Iterations | 100 |
| TabTransformer | Embedding dimension | 4, 8, 16, 32, 64, 128 |
| | Number of heads | UniformInt(1, 10) |
| | Number of attention blocks | UniformInt(1, 12) |
| | Attention dropout rate | Uniform(0, 0.5) |
| | Add norm dropout | Uniform(0, 0.5) |
| | Transformation activation | [Tanh, Relu, LeakyReLU] |
| | L2 | LogUniform(1e-6, 1e-1) |
| | Learning rate | LogUniform(1e-6, 1e-3) |
| | FF dropout | Uniform(0, 0.5) |
| | FF hidden multiplier | 1, 2, 3, 4, 5, 6, 7, 8, 9, 10 |
| | Out FF activation | [Tanh, Relu, LeakyReLU] |
| | Out FF dropout | Uniform(0, 0.5) |
| NODE | Learning rate | LogUniform(1e-5, 1) |
| | Number of layers | UniformInt(1, 10) |
| | Number of trees | UniformInt(2, 2048) |
| | Depth | UniformInt(1, 10) |
| Diet Network | Embedding choice | $X^T$, random |
| | Number of layers | UniformInt(1, 8) |
| | Hidden dimension | UniformInt(1, 512) |
| | Dropout | 0, Uniform([0,0.5]) |
| | Learning rate | LogUniform(1e-5, 1e-2) |
| | L2 | 0, LogUniform(1e-6, 1e-3) |

**Supplementary Table 2: Hyperparameter range for all baselines.**

## B    GRAPH CLASSIFICATION APPROACHES

Graph classification models are not relevant for PLATO's setting. In graph classification models, every input sample is a graph with node attributes, and a model must make a prediction for that graph. The PLATO problem setting breaks fundamental assumptions made by graph classification models, rendering them not applicable. First, graph classification models assume that different samples correspond to different graphs (Ying et al., 2021; Hu et al., 2020b;a). However, in PLATO every sample corresponds to the exact same graph. There is a single background knowledge graph for all samples, and every sample has input features that correspond to the exact same nodes within the knowledge graph. Second, graph classification approaches typically assume that every node in an input graph has a node attribute (Ying et al., 2021; Hu et al., 2020b;a). However, in PLATO only a small subset of the nodes in the knowledge graph have measured feature values. Finally, graph classification approaches typically assume small graphs: the largest graph classification task in the Open Graph Benchmark has only 244 nodes (Hu et al., 2020a). However in PLATO, the knowledge graph contains 108,447 and the smallest dataset has 12,932 features corresponding to nodes.

## C    RANK ORDERING OF METHODS FOR DATASETS WITH $d \gg n$

In Supplementary Table 3, we show the rank order performance of all models on all $d \gg n$ datasets. We find that PLATO exhibits consistent and strong performance while the performance of the baselines depends on the specific $d \gg n$ dataset. For example, TabTransformer is the second best performing of all models on the MNSCLC dataset but the worst performing of all models on the PDAC and CH datasets. Similarly, XGBoost is the second best performing of all models on PDAC but only the seventh best performing of all models on BRCA. The baselines with the most stable performance are LASSO and Ridge Regression which rank consistently between the second and fifth best of all models.

**Supplementary Table 3: For datasets with $d \gg n$, PLATO exhibits consistent and strong performance.** By contrast, the performance of the baselines varies with each dataset. For every dataset, the rank order of performance from Table 1 is shown. The best overall model is in **bold** and the second best model is underlined.

| Dataset | | MNSCLC | CM | PDAC | BRCA | CRC | CH |
|---|---|---|---|---|---|---|---|
| D/N | | 52.2 | 46.1 | 40.3 | 28.2 | 22.6 | 19.7 |
| Classic Stat ML | Ridge | 5 | 3 | 4 | 2 | 3 | 4 |
| Dim. Reduct. | PCA | 4 | 9 | 8 | 6 | 9 | 6 |
| Feat. Select. | LASSO | 3 | 2 | 3 | 5 | 2 | 3 |
| | STG | 6 | 4 | 7 | 4 | 7 | 5 |
| Decision Tree | XGBoost | 11 | 6 | 2 | 7 | 5 | 2 |
| Param. Pred. | Diet | 10 | 10 | 5 | 9 | 10 | 8 |
| Tabular DL | MLP | 7 | 5 | 6 | 8 | 4 | 9 |
| | NODE | 9 | 7 | 9 | 3 | 6 | 7 |
| | TabTransformer | 2 | 8 | 11 | 10 | 8 | 11 |
| | TabNet | 8 | 11 | 10 | 11 | 11 | 10 |
| Ours | PLATO | **1** | **1** | **1** | **1** | **1** | **1** |

**Supplementary Table 4: For datasets with $d \sim n$, PLATO is competitive with baselines.** For XGBoost is consistently the strongest baseline. For every dataset, the rank order of performance from Table 4 is shown. The best overall model is in **bold** and the second best model is underlined.

| Dataset | | ME | BC | SCLC | NSCLC |
|---|---|---|---|---|---|
| D/N | | 2.0 | 1.8 | 1.7 | 1.1 |
| Classic Stat ML | Ridge | 7 | 7 | 7 | 3 |
| Dim. Reduct. | PCA | 10 | 9 | 10 | 6 |
| Feat. Select. | LASSO | 6 | 4 | 4 | 7 |
| | STG | 4 | 3 | 5 | 5 |
| Decision Tree | XGBoost | 2 | 2 | 2 | **1** |
| Param. Pred. | Diet | 11 | 10 | 11 | 11 |
| Tabular DL | MLP | 8 | 6 | 8 | 8 |
| | NODE | 3 | 8 | 3 | 10 |
| | TabTransformer | 9 | 11 | 9 | 9 |
| | TabNet | 5 | 5 | 6 | 4 |
| Ours | PLATO | **1** | **1** | **1** | 2 |

# D  CODE DETAILS

Code to run PLATO will be included as a supplementary file in the final version of the manuscript.

# E  DATASET DETAILS

We compiled 6 datasets with $d \gg n$ and 4 datasets with $d \sim n$. In all datasets, a machine learning model must predict the response of a cell or mouse to a drug. In the tabular data, every row corresponds to a specific cell or mouse. Every column corresponds to a gene name. Every value corresponds to the amount of that gene in the tumor of the specific cell or mouse. The label is the response of the cell or mouse. All genes are nodes in the knowledge graph. In practice, the number of genes is large for all tasks and the number of samples is comparatively small making the drug response setting appropriate for $d \gg n$.

Dataset statistics, names, and sources are given below. Dataset pre-processing followed a standard process described in (Mourragui et al., 2021). Briefly, gene expression values underwent TMM normalization and log transformation (*i.e.* $\log(x + 1)$). Values were made to have zero mean and unit standard deviation. As labels, we used ln-ic50 for datasets from (Iorio et al., 2016; Yang et al., 2012; Garnett et al., 2012) and the minimum average percent tumor growth (*i.e.* "min-avg-pct-tumor-growth") for datasets from (Gao et al., 2015). For all datasets, we use a 200-dimensional ComplEx (Trouillon et al., 2016) embedding of the drug as the input feature vector. All datasets will be released in the final version of the manuscript.

| Dataset | Name | Source |
|---------|------|--------|
| MNSCLC | Non-small cell lung carcinoma | (Gao et al., 2015) |
| CM | Cutaneous melanoma | (Gao et al., 2015) |
| PDAC | Pancreatic ductal carcinoma | (Gao et al., 2015) |
| BRCA | Breast carcinoma | (Gao et al., 2015) |
| CRC | Colorectal cancer | (Gao et al., 2015) |
| CH | Chondrosarcoma | (Iorio et al., 2016; Yang et al., 2012; Garnett et al., 2012) |
| ME | Melanoma | (Iorio et al., 2016; Yang et al., 2012; Garnett et al., 2012) |
| BC | Breast carcinoma | (Iorio et al., 2016; Yang et al., 2012; Garnett et al., 2012) |
| SCLC | Small cell lung carcinoma | (Iorio et al., 2016; Yang et al., 2012; Garnett et al., 2012) |
| NSCLC | Non-small cell lung carcinoma | (Iorio et al., 2016; Yang et al., 2012; Garnett et al., 2012) |

## F  KNOWLEDGE GRAPH DETAILS

As a unified knowledge backbone for the datasets, we compile a general biomedical knowledge graph from prior studies (et al., 2020; 2016; Kuhn et al., 2015; Ruiz et al., 2021; Szklarczyk et al., 2020; Wishart et al., 2017a;b). A schematic of the KG is in Supplementary Figure 2. A detailed breakdown of relation types is in Supplementary Table 5.

Our knowledge graph contains 108,447 total nodes, including 7,975 drugs, 18,370 diseases, 11,447 phenotypes, 22,319 genes, 11,153 molecular functions, 28,748 biological processes, and 4,184 cellular components. Our knowledge graph contains 3,066,156 edges with 99 distinct relation types. All datasets include features which map to a subset of nodes in the knowledge graph, primarily genes and drugs. The remaining node types and their relationships serve as broader domain knowledge.

Edges between drug nodes and gene/protein nodes were derived from Drugbank (Wishart et al., 2017b), Gao (Gao et al., 2015), and the Genomics of Drug Sensitivity in Cancer (Yang et al., 2012; Iorio et al., 2016; Garnett et al., 2012). Edges between diseases and genes/proteins were derived from DisGeNet (Bauer-Mehren et al., 2010). Edges between diseases and phenotypes were derived from the Human Phenotype Ontology (et al., 2016). Edges between drugs and diseases were derived from the Multiscale Interactome (Ruiz et al., 2021). Edges between drugs and side effects were derived from SIDER (Kuhn et al., 2015). Edges between genes/proteins and other genes/proteins were derived from BioGRID (Oughtred et al., 2019), (Rual et al., 2005), the Database of Interacting Proteins (Salwinski et al., 2004), (et al., 2020), (Menche et al., 2015), (Rolland et al., 2014), (Yu et al., 2011), (Venkatesan et al., 2009), and STRING (Szklarczyk et al., 2020). Finally, edges from genes/proteins to molecular functions, biological processes, and cellular components as well as edges between molecular functions, biological processes, and cellular components were derived from the Gene Ontology (Consortium, 2018). The full knowledge graph will be included as a supplementary file in the final version of the manuscript.

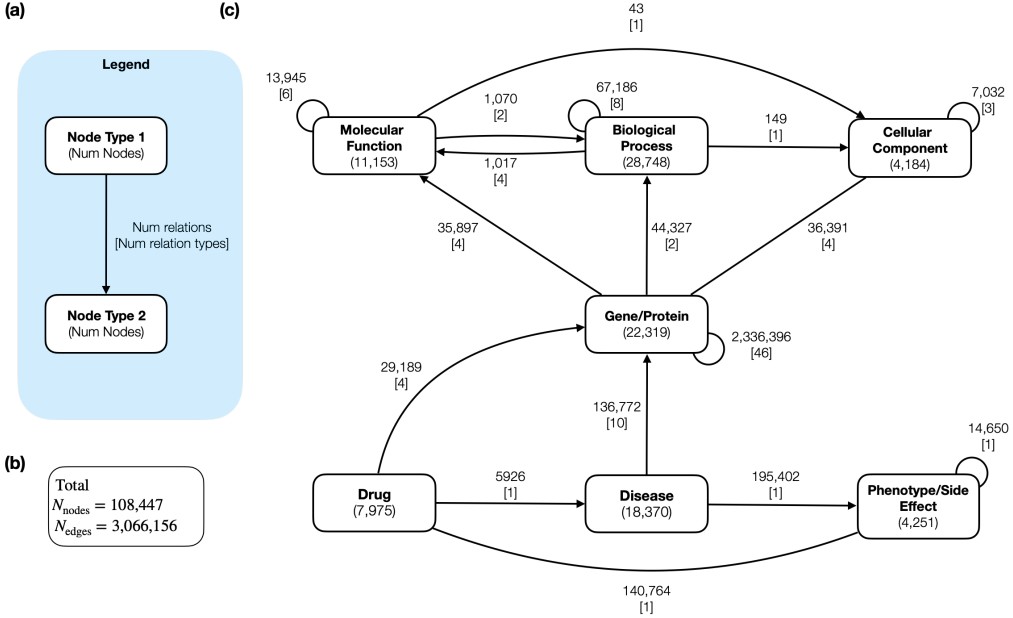

**Supplementary Figure 2: Knowledge graph as a unified knowledge backbone.** We constructed a knowledge graph as a unified knowledge backbone across all 6 datasets. (a) Legend. For each node type, the number of nodes is given in parentheses. Between node types, the number of edges and the number of relation types are given. (b) Number of total nodes and edges across entire knowledge graph. (c) Visual schematic of knowledge graph across each node type.

| Head type | Relation | Tail type | # edges |
|---|---|---|---|
| BiologicalProcess | EndsDuring | BiologicalProcess | 1 |
| BiologicalProcess | HappensDuring | BiologicalProcess | 8 |
| BiologicalProcess | HasPart | BiologicalProcess | 229 |
| BiologicalProcess | IsA | BiologicalProcess | 53015 |
| BiologicalProcess | NegativelyRegulates | BiologicalProcess | 2768 |
| BiologicalProcess | PartOf | BiologicalProcess | 5193 |
| BiologicalProcess | PositivelyRegulates | BiologicalProcess | 2756 |
| BiologicalProcess | Regulates | BiologicalProcess | 3216 |
| BiologicalProcess | OccursIn | CellularComponent | 149 |
| BiologicalProcess | HasPart | MolecularFunction | 173 |
| BiologicalProcess | NegativelyRegulates | MolecularFunction | 269 |
| BiologicalProcess | PositivelyRegulates | MolecularFunction | 274 |
| BiologicalProcess | Regulates | MolecularFunction | 301 |
| CellularComponent | HasPart | CellularComponent | 179 |
| CellularComponent | IsA | CellularComponent | 4863 |
| CellularComponent | PartOf | CellularComponent | 1990 |
| Disease | AlteredExpression | Gene | 7157 |
| Disease | Biomarker | Gene | 107160 |
| Disease | ChromosomalRearrangement | Gene | 162 |
| Disease | FusionGene | Gene | 166 |
| Disease | GeneticVariation | Gene | 15076 |
| Disease | GermlineCausalMutation | Gene | 4677 |
| Disease | ModifyingMutation | Gene | 10 |
| Disease | SomaticCausalMutation | Gene | 130 |
| Disease | SusceptibilityMutation | Gene | 441 |
| Disease | Therapeutic | Gene | 1793 |
| Disease | Has | Phenotype | 195402 |
| Drug | Treats | Disease | 5926 |
| Drug | Carries | Gene | 866 |
| Drug | Enzymes | Gene | 5382 |
| Drug | Targets | Gene | 19817 |
| Drug | Transports | Gene | 3124 |
| Drug | Has | Phenotype | 140764 |
| Gene | Associates | BiologicalProcess | 43857 |
| Gene | NotAssociates | BiologicalProcess | 470 |
| Gene | Associates | CellularComponent | 35306 |
| Gene | Colocalizes | CellularComponent | 914 |
| Gene | NotAssociates | CellularComponent | 160 |
| Gene | NotColocalizes | CellularComponent | 11 |
| Gene | Acetylation | Gene | 9 |
| Gene | Activation | Gene | 58502 |
| Gene | AdpRibosylation | Gene | 2 |
| Gene | Ampylation | Gene | 5 |
| Gene | Association | Gene | 18 |
| Gene | Binary | Gene | 56565 |
| Gene | Binding | Gene | 287641 |
| Gene | Catalysis | Gene | 344801 |
| Gene | Cleavage | Gene | 22 |
| Gene | Complexes | Gene | 62552 |
| Gene | CovalentBinding | Gene | 52 |
| Gene | Deacetylation | Gene | 8 |
| Gene | Demethylation | Gene | 6 |
| Gene | Dephosphorylation | Gene | 26 |

| | | | |
|---|---|---|---|
| Gene | Deubiquitination | Gene | 18 |
| Gene | DirectInteraction | Gene | 2904 |
| Gene | DisulfideBond | Gene | 5 |
| Gene | DosageGrowthDefect | Gene | 9 |
| Gene | DosageLethality | Gene | 112 |
| Gene | DosageRescue | Gene | 63 |
| Gene | Enzymatic | Gene | 2 |
| Gene | Expression | Gene | 188 |
| Gene | GeneticInterference | Gene | 32 |
| Gene | Hydroxylation | Gene | 26 |
| Gene | Inhibition | Gene | 20108 |
| Gene | Kinase | Gene | 11960 |
| Gene | Literature | Gene | 174162 |
| Gene | Metabolic | Gene | 10646 |
| Gene | Methylation | Gene | 25 |
| Gene | NegativeGenetic | Gene | 3449 |
| Gene | OxidoreductaseActivityElectronTransferAssay | Gene | 2 |
| Gene | PhenotypicEnhancement | Gene | 209 |
| Gene | PhenotypicSuppression | Gene | 214 |
| Gene | Phosphorylation | Gene | 166 |
| Gene | Phosphotransfer | Gene | 1 |
| Gene | PhysicalAssociation | Gene | 824164 |
| Gene | PositiveGenetic | Gene | 2331 |
| Gene | PostTranslationalModification | Gene | 5306 |
| Gene | ProteinCleavage | Gene | 48 |
| Gene | PutativeSelfInteraction | Gene | 3 |
| Gene | Reaction | Gene | 400658 |
| Gene | Regulation | Gene | 2650 |
| Gene | Signaling | Gene | 65412 |
| Gene | SyntheticGrowthDefect | Gene | 407 |
| Gene | SyntheticLethality | Gene | 816 |
| Gene | SyntheticRescue | Gene | 91 |
| Gene | Associates | MolecularFunction | 35012 |
| Gene | Contributes | MolecularFunction | 596 |
| Gene | NotAssociates | MolecularFunction | 285 |
| Gene | NotContributes | MolecularFunction | 4 |
| MolecularFunction | PartOf | BiologicalProcess | 1068 |
| MolecularFunction | Regulates | BiologicalProcess | 2 |
| MolecularFunction | OccursIn | CellularComponent | 43 |
| MolecularFunction | HasPart | MolecularFunction | 204 |
| MolecularFunction | IsA | MolecularFunction | 13631 |
| MolecularFunction | NegativelyRegulates | MolecularFunction | 42 |
| MolecularFunction | PartOf | MolecularFunction | 11 |
| MolecularFunction | PositivelyRegulates | MolecularFunction | 27 |
| MolecularFunction | Regulates | MolecularFunction | 30 |
| Phenotype | IsA | Phenotype | 14650 |

**Supplementary Table 5: Knowledge graph relations between node types.**

