# OpenReview forum: "Tabular Deep Learning when $d \gg n$ by Using an Auxiliary Knowledge Graph"
_ICLR.cc/2023/Conference — Submitted to ICLR 2023_

### Official Review · Reviewer_Ddzi · 2022-10-24

**Confidence:** 4
**Clarity, Quality, Novelty And Reproducibility:** <mentioned in comments above.>
**Correctness:** 2
**Technical Novelty And Significance:** 2
**Empirical Novelty And Significance:** 1
**Recommendation:** 1

**Strength And Weaknesses:**


Pros:
1. Figure 1 explains their methodology in a good and succinct manner.

Comments:
1. It will be helpful to cover some related graph recovery methods to get auxiliary graphs for more adoption.
2. (I) In sec 3.2, the rationale given to share the parameters of the first layer does not seem principled to me.  “Typically this intuition is missed since T learns the parameters Θ [1] j and Θ [1] k associated with two features j and k independently by gradient backpropagation.” The features do have common parameters in the subsequent layers through which their commonalities can be modelled.
3. This methodology seems to learn a projection of the input data to another space, say R^h (here referred to as layer 1 of the MLP). The method comes up with `d’ vectors in R^h space that are put together as a matrix to obtain the $\Theta[1]$. This is done by running message-passing on the KG which takes individual feature inputs as one-hot vectors to obtain a vector in R^h space.
4. How does it perform for out-of-distribution samples?
5. Please share details about any ablation study done, which empirically shows that having the first layer obtained from this method works better than just training a simple MLP. What happens when there are more than 3-4 layers in MLP and what if there are no learnable MLP layer and just this direct projection layer?

I have some major concerns about this methodology (pts 2&3 in comments). The rationale given by the authors seems to be hand-wavy and not grounded. I hereby request the authors to develop a mathematically rigorous approach to support their claims.


**Summary Of The Paper:**

This work attempts to address the curse of high dimensionality problem for the tabular data. The authors build their model, PLATO, on the well formed insight that auxiliary KG about the feature connections can improve performance as the sparsity induced in the model helps with the generalizations in the cases where d>>n. Their method predicts the first layer weights of the MLP and then the remaining layers of the MLP are trained using the loss function. They evaluate their method on different datasets and report improved performance.

**Summary Of The Review:**

I have some major concerns about this methodology (pts 2&3 in comments). The rationale given by the authors seems to be hand-wavy and not grounded. I hereby request the authors to develop a mathematically rigorous approach to support their claims.

---

> ### Author Response · Authors · 2022-11-05
> **Clarify specific graph recovery papers? Thanks for the review!**
>
> > **It will be helpful to cover some related graph recovery methods to get auxiliary graphs for more adoption.**
>
> Could the reviewer please suggest the specific graph recovery papers they view as relevant? PLATO is not a graph recovery method. PLATO assumes that an auxiliary KG already exists and uses this KG to improve prediction on a tabular dataset.

---

> > ### Author Response · Authors · 2022-11-15
> > **We address this comment in "Response to Ddzi (Part 1)" above!**
> >
> > We address this comment in "Response to Ddzi (Part 1)" above! Thank you!

---

> ### Author Response · Authors · 2022-11-05
> **Clarify concern with method step (3)? Thanks for the review!**
>
> > **3. This methodology seems to learn a projection of the input data to another space, say R^h (here referred to as layer 1 of the MLP). The method comes up with `d’ vectors in R^h space that are put together as a matrix to obtain the Theta[1]. This is done by running message-passing on the KG which takes individual feature inputs as one-hot vectors to obtain a vector in R^h space.**
>
> The reviewer says that comment (3) is a major concern. However, the reviewer does not explain what the concern is beyond describing a specific step of the method. Could the reviewer please clarify what their concern is about this step?

---

> > ### Comment · Reviewer_Ddzi · 2022-11-05
> > **Reply to authors for method step (3)**
> >
> > Thank you for looking into the method step feedback. I am not able to follow what does this step achieves and how does it contribute towards the entire architecture. It will be great to see some mathematical justification here.

---

> > > ### Author Response · Authors · 2022-11-15
> > > **We address this comment in "Response to Ddzi (Part 4)" above!**
> > >
> > > We address this comment in "Response to Ddzi (Part 4)" above! Thank you!

---

> ### Author Response · Authors · 2022-11-14
> **Response to Ddzi (Part 1)**
>
> > **Concise summary and context for response**
>
> We thank the reviewer for their questions! We would first like to provide a concise summary to clarify PLATO’s methodology and provide context for our response. We paraphrase Reviewer AKNa’s excellent overview below.
>
> **PLATO is a MLP where the weights in the first layer are predicted from an auxiliary KG with input features as nodes as follows:**
> 1. A c-dimensional embedding $ \mathbf{M}_j \in \mathbb{R}^c $ is learned for every input feature $ j $ via pre-training on the auxiliary KG (no tabular data is used).
> 2. A message passing algorithm that uses the knowledge graph is used to learn another embedding $ \mathbf{Q}_j \in \mathbb{R}^c$ for each input feature $ j $ based on its neighbors. In each round of message passing, the embedding of a feature is a weighted combination of its embedding in the previous round and all its neighbors in the previous round.
> 3. A small neural network $ \mathcal{P} $ that is shared across all input features takes in each feature embedding $ \mathbf{Q}_j $ and produces the weights in the first layer of the MLP that correspond to that feature $ \mathbf{\hat{\Theta}}^{[1]}_j \in \mathbb{R}^h$ where $ h $ is the number of hidden units. The weights in the remaining layers of the MLP are learned normally.
> 4. Finally, the full input sample $ \mathbf{X}_i$ is fed into the MLP which makes a prediction $ \mathbf{\hat{y}}_i$.
>
> ---
>
> > **"The authors build their model, PLATO, on the well formed insight that auxiliary KG about the feature connections can improve performance... They evaluate their method on different datasets and report improved performance... Figure 1 explains their methodology in a good and succinct manner."**
>
> We thank the reviewer for pointing out the "well formed insight" that underlies PLATO's method, the many datasets on which PLATO improves performance, and the clarity of Figure 1 which describes the method!
>
> ---
>
> > **1. It will be helpful to cover some related graph recovery methods to get auxiliary graphs for more adoption.**
>
> **Expanding PLATO to problem settings in which an auxiliary knowledge graph does not already exist but is instead constructed from the tabular data or adjacent unlabeled data is an exciting area of future work for PLATO!** PLATO’s problem setting currently assumes that an auxiliary knowledge graph already exists (Section 3.1). In response to the reviewer, we will add the following sentence to the discussion and reference related work in the revised version of the manuscript:
>
> “Finally, PLATO depends on the existence of an auxiliary KG though future work may leverage existing methods to construct the KG from the data itself (Qiao et al., 2018) or auxiliary unlabeled data (Chong et al., 2020).”
>
> References:
>
> [1] Lishan Qiao, Limei Zhang, Songcan Chen, and Dinggang Shen. Data-driven graph construction and graph learning: A review. Neurocomputing, 312:336–351, 2018.
>
> [2] Yanwen Chong, Yun Ding, Qing Yan, and Shaoming Pan. Graph-based semi-supervised learning: A review. Neurocomputing, 408:216–230, 2020
>
> ---

---

> > ### Author Response · Authors · 2022-11-14
> > **Response to Ddzi (Part 2)**
> >
> > > **Can the authors mathematically justify the methodology, especially the parameter prediction (part 2 in the comments) and the message-passing (part 3 in the comments)?**
> >
> > PLATO is a deep learning method for tabular data with $d \gg n$ which uses an auxiliary knowledge graph containing input features as nodes. **The key mathematical question that PLATO solves is how to reduce the number of trainable parameters in the first layer of a MLP.** For a tabular dataset with $ d $ features, the first layer of a MLP with $ h $ hidden units has $\mathbf{\Theta}^{[1]} \in \mathbb{R}^{d \times h}$ trainable parameters. For high dimensional datasets with few labeled samples (i.e. $ d \gg n $), the large number of trainable parameters in the first layer of the MLP poses a challenge. The PLATO algorithm is designed to predict the parameters $ \mathbf{\hat{\Theta}}^{[1]} \in \mathbb{R}^{d \times h}$ in the first layer of the MLP using a smaller number of trainable parameters, ultimately enabling effective predictions when $d \gg n$.
> >
> > Methodologically, PLATO enables the prediction of the parameters in the first layer of the MLP by using an auxiliary KG which includes the input features as nodes. We observe that every trainable parameter in the first layer of a MLP is associated with an input feature. For example, input feature $j$ has a vector $\mathbf{\Theta}^{[1]}_j \in \mathbb{R}^{h}$ of associated trainable parameters (Figure 1e). To predict these parameters, PLATO first learns a compact embedding $ \mathbf{Q}_j \in \mathbb{R}^c$ for every input feature $ j $ from the auxiliary KG. PLATO then uses a neural network $ \mathcal{P} $ which is shared across all features to predict the parameters in the first layer of the MLP according to $\mathbf{\hat{\Theta}}^{[1]}_j = \mathcal{P}(\mathbf{Q}_j; \mathbf{\Phi})$. $ \mathbf{\Phi} $ are the trainable parameters of $ \mathcal{P} $.
> >
> > **PLATO’s approach ultimately reduces the number of trainable parameters in the first layer of the MLP. First, note that the number of trainable parameters in $\mathcal{P}$ is small compared to the number of trainable parameters $dh$ that would typically be in the first layer of the MLP.** $\mathcal{P}$ need only transform every $\mathbf{Q}_j \in \mathbb{R}^c$ to $\mathbf{\hat{\Theta}} \in \mathbb{R}^h$. If we assume that $\mathcal{P}$ is a single-layer neural network, then $ |\mathbf{\Phi}| = ch $. The dimensionality of the feature embedding, $ c $, is a hyperparameter and is much less than the number of input features $ d $. As a result, $ |\Phi| = ch \ll dh$ and $ \mathcal{P}$ has fewer trainable parameters than would typically be in the first layer of the MLP.
> >
> > **Moreover, note that PLATO learns the embedding $ \mathbf{Q}_j $ with very few trainable parameters from the point-of-view of the tabular data.** To produce $ \mathbf{Q}_j \in \mathbb{R}^c $, PLATO first pre-trains $ \mathbf{M}_j \in \mathbb{R}^c $, an embedding for every input feature, by using a self-supervision objective on the KG. $ \mathbf{M}_j $ is fixed after pre-training and thus does not introduce any trainable parameters from the point-of-view of the tabular dataset. PLATO then updates $\mathbf{M}_j$ to $\mathbf{Q}_j$, a new embedding for every input feature, by using message-passing on the auxiliary KG. In each round of message passing, the embedding of a feature is a weighted sum of its embedding in the previous round and all its neighbors in the previous round. The weights in the weighted sum are predicted from a small neural network $ \mathcal{A} $ with trainable parameters $ \mathbf{\Pi}$. For a sample $ i $ and input features $j$ and $k$, the neural network $ \mathcal{A} $ takes in the feature values for a sample $ \mathbf{X}_i\mathbf{}_j $ and $ \mathbf{X}_i\mathbf{}_k$ and outputs a scalar weight. The number of trainable parameters in $ \mathbf{\Pi}$ is small since the input of $ \mathcal{A} $ is $ \mathbb{R}^2 $ and the output of $ \mathcal{A}$ is $ \mathbb{R} $. Moreover, $ \mathcal{A}$ and its parameters $ \mathbf{\Pi}$ are shared for all samples and features. Therefore, $ |\mathbf{\Pi}| $ is small , and the computation of $ \mathbf{Q}_j $ thus introduces a minimal number of trainable parameters.
> >
> > **Ultimately, PLATO reduces the number of trainable parameters in the first layer of a MLP by predicting these parameters from an auxiliary KG that includes input features as nodes.** PLATO replaces $ dh $ trainable parameters in the first layer of a MLP with trainable parameters $ \mathbf{\Phi} $ and $ \mathbf{\Pi} $ which are small in comparison. Ultimately, PLATO outperforms ten statistical and deep baselines on 6 datasets with $ d \gg n$, achieving performance improvements up to +10.19% (Table 2).
> >
> > ---

---

> > > ### Author Response · Authors · 2022-11-14
> > > **Response to Ddzi (Part 3)**
> > >
> > > > **2. "In sec 3.2, the rationale given to share the parameters of the first layer does not seem principled to me. “Typically this intuition is missed since T learns the parameters Θ [1] j and Θ [1] k associated with two features j and k independently by gradient backpropagation.” The features do have common parameters in the subsequent layers through which their commonalities can be modeled."**
> > >
> > > **We thank the reviewer for their comment and have removed this sentence from Section 3.2. The key mathematical question that PLATO solves is how to reduce the trainable parameters in the first layer of a MLP.** An intuition that PLATO uses as a component of solving that question is that if two input features $ j $ and $ k $ are related, then the trainable parameters in the first layer of the MLP that correspond to those features $ \mathbf{\Theta}^{[1]}_j $ and $ \mathbf{\Theta}^{[1]}_k $ are likely to be related too. Such an intuition motivates our use of a shared neural network $ \mathcal{P} $ to predict the parameters in the first layer of the MLP from prior information known about each input feature.
> > >
> > > Our intuition draws additional support from the graph regularization literature, in which weights of a predictive model that correspond to adjacent nodes in a graph are regularized to be similar [1-3].
> > >
> > > References:
> > >
> > > [1] Grosenick, Logan, et al. "Interpretable whole-brain prediction analysis with GraphNet." NeuroImage 72 (2013): 304-321.
> > >
> > > [2] Li, Caiyan, and Hongzhe Li. "Network-constrained regularization and variable selection for analysis of genomic data." Bioinformatics 24.9 (2008): 1175-1182.
> > >
> > > [3] Hallac, David, Jure Leskovec, and Stephen Boyd. "Network lasso: Clustering and optimization in large graphs." Proceedings of the 21th ACM SIGKDD international conference on knowledge discovery and data mining. 2015.
> > >
> > > ---

---

> > > > ### Author Response · Authors · 2022-11-14
> > > > **Response to Ddzi (Part 4)**
> > > >
> > > > > **3. "This methodology seems to learn a projection of the input data to another space, say R^h (here referred to as layer 1 of the MLP). The method comes up with `d’ vectors in R^h space that are put together as a matrix to obtain the Theta[1]. This is done by running message-passing on the KG which takes individual feature inputs as one-hot vectors to obtain a vector in R^h space."**
> > > >
> > > > We thank the reviewer for their question! Here, the reviewer asks us to clarify the message-passing step of the PLATO architecture. For context, recall that PLATO predicts the parameters in the first layer of a MLP. For every input feature $ j $, PLATO predicts $ \mathbf{\hat{\Theta}}_j^{[1]} \in \mathbb{R}^h $, the parameters in the first layer of the MLP with $ h $ hidden units that correspond to that feature. PLATO predicts $ \mathbf{\hat{\Theta}}_j^{[1]}$ from $ \mathbf{Q}_j \in \mathbb{R}^c$, an embedding that is learned for that feature $ j $. PLATO repeats this for all $d$ features to predict $\mathbf{\hat{\Theta}}^{[1]} \in \mathbb{R}^{d \times h}$ the parameters in the first layer of the MLP.
> > > >
> > > > **During the message-passing step, PLATO produces the feature embedding** $ \mathbf{Q}_j $ by starting with a pre-trained embedding for every input feature $\mathbf{M}_j \in \mathbb{R}^c$ (Methods 3.3.1). Then in each round of message passing, the embedding of a feature is a weighted sum of its embedding in the previous round and all its neighbors in the previous round. The weights in the weighted sum are predicted from a small neural network $ \mathcal{A} $ with trainable parameters $ \mathbf{\Pi}$. For a sample $ i $ and input features $j$ and $k$, the neural network $ \mathcal{A} $ takes in the feature values for a sample $ \mathbf{X}_i\mathbf{}_j $ and $ \mathbf{X}_i\mathbf{}_k$ and outputs a scalar value that weights the prior embeddings of features $ j $ and $ k $ in the weighted sum. Ultimately, message-passing produces the feature embedding $ \mathbf{Q}_j $.
> > > >
> > > > The message-passing step has two key motivations. **First, updating the embedding of an input feature based on the embedding of its neighbors allows us to capture the principle of network homophily.** Homophily, the principle that nodes that share edges in a network tend to have similar characteristics, has been established for a range of networks [1-2], and is a key component of graph neural network approaches [3]. Message passing allows PLATO to update the embedding of a given node to consider information from its neighboring nodes, capturing homophily.
> > > >
> > > > **The second motivation of message passing is to allow the parameters in the first layer of the MLP to vary for every input sample $ \mathbf{X}_i$, thereby increasing the representational capacity of the MLP.** Since $\mathbf{Q}_j$ depends on $ \mathbf{X}_i$ and the parameters in the first layer of the MLP are predicted from $ \mathbf{Q}_j$, the parameters in the first layer of the MLP also vary for every $ \mathbf{X}_i$. Allowing the parameters in the first layer of a MLP to vary with the input sample is used in the dynamic neural network literature to increase representational power [4]. The specific message-passing architecture we use allows $ \mathbf{Q}_j$ to depends on $ \mathbf{X}_i$ in a way that minimizes the number of additional trainable parameters that are introduced (see Response to Ddzi Part 2 for further discussion on the number of trainable parameters).
> > > >
> > > > **Finally, we provide empirical proof of the necessity of the message-passing step.** Table 2 and the second Results subsection demonstrates that PLATO with message-passing outperforms PLATO without message-passing (0.583±0.019 vs. 0.522±0.030). Message-passing is thus a critical step of the PLATO architecture.
> > > >
> > > > References:
> > > >
> > > > [1] McPherson, Miller, Lynn Smith-Lovin, and James M. Cook. "Birds of a feather: Homophily in social networks." Annual review of sociology (2001): 415-444.
> > > >
> > > > [2] Gerber, Elisabeth R., Adam Douglas Henry, and Mark Lubell. "Political homophily and collaboration in regional planning networks." American Journal of Political Science 57.3 (2013): 598-610.
> > > >
> > > > [3] Zhu, Jiong, et al. "Beyond homophily in graph neural networks: Current limitations and effective designs." Advances in Neural Information Processing Systems 33 (2020): 7793-7804.
> > > >
> > > > [4] Han, Yizeng, et al. "Dynamic neural networks: A survey." IEEE Transactions on Pattern Analysis and Machine Intelligence (2021).
> > > >
> > > > ---

---

> > > > > ### Author Response · Authors · 2022-11-14
> > > > > **Response to Ddzi (Part 5)**
> > > > >
> > > > > > **4. How does PLATO perform for out-of-distribution samples?**
> > > > >
> > > > > **Examining the performance of PLATO on out-of-distribution samples is an exciting avenue for future work!** The current manuscript does not make any claims about out-of-distribution samples. Indeed, we follow the rigorous data splitting and evaluation methodology used in recent tabular data benchmarks to evaluate PLATO [1, 2]. These benchmarks do not contain out-of-distribution testing for tabular models.
> > > > >
> > > > > References:
> > > > >
> > > > > [1] Grinsztajn, Leo, Edouard Oyallon, and Gael Varoquaux. "Why do tree-based models still outperform deep learning on typical tabular data?." NeurIPS Datasets and Benchmarks Track. 2022.
> > > > >
> > > > > [2] Yury Gorishniy, Ivan Rubachev, Valentin Khrulkov, and Artem Babenko. Revisiting deep learning models for tabular data. NeurIPS, 34, 2021.
> > > > >
> > > > > ---
> > > > >
> > > > > > **5a. Please share details about any ablation study done, which empirically shows that having the first layer obtained from this method works better than just training a simple MLP. What happens when there are more than 3-4 layers in MLP?**
> > > > >
> > > > > **PLATO outperforms a simple MLP, including one with more than 3-4 layers, on all 6 datasets with $d \gg n$.** Table 2 already demonstrates the effectiveness of PLATO versus such a simple MLP. We include the relevant rows again here for convenience.
> > > > >
> > > > > |       | MNSCLC          | CM              | PDAC            | BRCA            | CRC             | CH              |
> > > > > |-------|-----------------|-----------------|-----------------|-----------------|-----------------|-----------------|
> > > > > | MLP   | 0.128 +/- 0.126 | 0.322 +/- 0.043 | 0.289 +/- 0.047 | 0.240 +/- 0.067 | 0.355 +/- 0.022 | 0.044 +/- 0.039 |
> > > > > | PLATO | 0.272 +/- 0.130 | 0.435 +/- 0.022 | 0.400 +/- 0.021 | 0.583 +/- 0.019 | 0.401 +/- 0.019 | 0.770 +/- 0.003 |
> > > > >
> > > > >
> > > > > Note that PLATO outperforms a simple MLP, even when there are more than 3-4 layers in the MLP (Table 1, Appendix B). Each MLP performance value above was optimized using a hyperparameter sweep on each dataset with 500 configurations following best practices from recent tabular data benchmarks [1, 2]. Already, we considered MLP configurations with more than 3-4 layers, indicating that PLATO substantially outperforms MLPs with more than 3-4 layers. The relevant part of Appendix B is below for convenience.
> > > > >
> > > > > | **MLP Configuration** | Range                     |
> > > > > |-----------------------|---------------------------|
> > > > > | Number of layers      | UniformInt(1, 8)          |
> > > > > | Hidden dimension      | UniformInt(1, 512)        |
> > > > > | Dropout               | 0, Uniform([0, 0.5])      |
> > > > > | Learning rate         | LogUniform(1e-5, 1e-2)    |
> > > > > | L2                    | 0, LogUniform(1e-6, 1e-3) |
> > > > >
> > > > > References:
> > > > >
> > > > > [1] Grinsztajn, Leo, Edouard Oyallon, and Gael Varoquaux. "Why do tree-based models still outperform deep learning on typical tabular data?." NeurIPS Datasets and Benchmarks Track. 2022.
> > > > >
> > > > > [2] Yury Gorishniy, Ivan Rubachev, Valentin Khrulkov, and Artem Babenko. Revisiting deep learning models for tabular data. NeurIPS, 34, 2021.
> > > > >
> > > > > ---
> > > > >
> > > > > > **5b. What if there are no learnable MLP layers and just the direct projection layer (i.e. PLATO makes the prediction directly after the parameter prediction layer)?**
> > > > >
> > > > > We thank the reviewer for their excellent question! We are currently conducting a new ablation study in which the PLATO MLP has a single layer and the parameters in that layer are predicted from the auxiliary KG. We will update this comment with the experimental results as soon as they are available!
> > > > >
> > > > > ---

---

> > > > > > ### Author Response · Authors · 2022-11-17
> > > > > > **Thank you!**
> > > > > >
> > > > > > We thank the reviewer for their time and thoughtful questions! If the reviewer has any further concerns, please let us know! If not, we would appreciate the reviewer considering raising their score. Thank you!

---

> > > > > > > ### Author Response · Authors · 2022-12-13
> > > > > > > **Final commentary**
> > > > > > >
> > > > > > > If the reviewer has had the chance to read our response, we would appreciate any final commentary! If there are no further concerns, we would appreciate the reviewer considering raising their score. Thank you!

---

> > > > > > ### Author Response · Authors · 2022-11-19
> > > > > > **Response to Ddzi, Experiment for 5b**
> > > > > >
> > > > > > > **5b. What if there are no learnable MLP layers and just the direct projection layer (i.e. PLATO makes the prediction directly after the parameter prediction layer)?**
> > > > > >
> > > > > > We thank the reviewer for their excellent question! We conduct a **new experiment demonstrating that PLATO with learnable MLP layers after the first layer outperforms an ablation of PLATO with a single MLP layer whose parameters are predicted!**
> > > > > >
> > > > > > As the reviewer suggests, we construct a modified version of PLATO in which PLATO predicts the parameters in the first layer of a MLP and that first layer directly predicts the label. This is equivalent to a linear regression where the weights in the linear regression are predicted by PLATO. By contrast, the standard configuration of PLATO predicts the weights in the first layer of a MLP with $h$ hidden units. The first layer is then followed by additional MLP layers whose weights are learned normally. The final layer of the MLP predicts the label.
> > > > > >
> > > > > > PLATO’s standard configuration outperforms the ablated version of PLATO with just a single MLP layer whose parameters are predicted.
> > > > > >
> > > > > > | Model                              | PearsonR (Test) on BRCA |
> > > > > > |------------------------------------|-------------------------|
> > > > > > | PLATO with learnable MLP layers    | 0.583 +/- 0.019         |
> > > > > > | PLATO with single, predicted layer | 0.550 +/- 0.020         |

---

### Official Review · Reviewer_yX2X · 2022-10-24

**Confidence:** 3
**Correctness:** 3
**Technical Novelty And Significance:** 1
**Empirical Novelty And Significance:** 2
**Recommendation:** 3

**Clarity, Quality, Novelty And Reproducibility:**


The paper is reasonably clear but not stellar. In particular it does not explain how the specificities of the method where obtained.

The novelty is limited given the huge amount of papers on graph regularization.

Given that no code is available, the reproducibility is limited.


**Strength And Weaknesses:**

The idea of using a knowledge graph to build a graph neural net on the features is interesting.

However, this paper seems more targetted as computational biology applications, and not tabular data in general. Indeed, outside computational biology, it seems quite unlikely to find the configuration of many columns all corresponding to entries of a knowledge base. Can this paper be applied outside computational biology? If so, it would be interesting to demonstrate it.


Did the author compare to classic graph-regularized models? There is a literature that is more than 10 years old on the topic, including in computational biologie.

I do not understand where things such as the line 3 of algorithm 1 come from. I just do not understand how was the algorithm designed. By trial and error?

How are the error bars computed on the tables? Given the small number of n, I would expect much larger error bars.

PCA is not a supervised method. How can it be used in the experimental tables?



**Summary Of The Paper:**

This submission contributes a learning method for high-dimensional data that uses an associated knowledge graph between the features to create a lower-dimension representation. The methods pretrains a data-transformation mechanism on the knowledge graph, creates attention weights, and used message passing on the corresponding graph neural network to output an embedding of input data, which consistutes the intermediate layer of an MLP. The method is benchmarked on 6 biomedical datasets, with a dozen thousands features and hundreds samples, where it outperforms classic methods.


**Summary Of The Review:**

There is an interesting idea, but this seems more like a computational biology paper than a general tabular learning but. The paper does not contribute understandible and insight on top of the graph regularization work.

---

> ### Author Response · Authors · 2022-11-05
> **Clarify specific graph regularization models? Thanks for the review!**
>
>
>
> > **Did the author compare to classic graph-regularized models?**
>
> Could the reviewer please clarify which graph-regularized models they are referring to?

---

> > ### Comment · Reviewer_yX2X · 2022-11-05
> > **There is a lot of work using graphs as a structured regularizer**
> >
> > > Could the reviewer please clarify which graph-regularized models they are referring to?
> >
> > Many many methods have been developed using a graph structure in the regularization. For instance, GraphNet (Grosenick et al 2013) and SConES (Azencott et al 2013) come to my mind, both are fairly well cited. I know that there are other approaches in this line. I'm an author of none of these approaches, but I still see it as quite frustrating that none of this body of literature is acknowledged.

---

> > > ### Author Response · Authors · 2022-11-14
> > > **We address this comment in "Response to yX2X (Part 2)" above!**
> > >
> > > We address this comment in "Response to yX2X (Part 2)" above! Thank you!

---

> ### Author Response · Authors · 2022-11-14
> **Response to yX2X (Part 1)**
>
> > **1. The reviewer’s summary describes PLATO as a dimensionality reduction approach for input data.**
>
> We thank the reviewer for their thoughtful questions! We would first like to provide a concise summary to clarify PLATO’s methodology and provide context for our response. For clarification, we would like to point out that PLATO does not create a lower-dimensional representation of the input sample with the knowledge graph. Instead, the knowledge graph is being used to predict the weights in the first layer of a MLP. We paraphrase Reviewer AKNa’s excellent overview below.
>
> **PLATO is a MLP where the weights in the first layer are predicted from an auxiliary KG with input features as nodes as follows:**
>
> 1. A c-dimensional embedding $ \mathbf{M}_j \in \mathbb{R}^c $ is learned for every input feature $ j $ via pre-training on the auxiliary KG (no tabular data is used).
> 2. A message passing algorithm that uses the knowledge graph is used to learn another embedding $ \mathbf{Q}_j \in \mathbb{R}^c$ for each input feature $ j $ based on its neighbors. In each round of message passing, the embedding of a feature is a weighted combination of its embedding in the previous round and all its neighbors in the previous round.
> 3. A small neural network $ \mathcal{P} $ that is shared across all input features takes in each feature embedding $ \mathbf{Q}_j $ and produces the weights in the first layer of the MLP that correspond to that feature $ \mathbf{\hat{\Theta}}^{[1]}_j \in \mathbb{R}^h$ where $ h $ is the number of hidden units.
> 4. Finally, the full input sample $ \mathbf{X}_i$ is fed into the MLP which makes a prediction $ \mathbf{\hat{y}}_i$.
>
> ---
>
> > **2. The idea of using a knowledge graph to build a graph neural net on the features is interesting.**
>
> We thank the reviewer for pointing out that our methodology is “interesting”!
>
> ---
>
> > **3. Does PLATO apply to general tabular datasets? Can PLATO be applied outside of computational biology?**
>
> **The PLATO method is not limited to computational biology and can be applied to any tabular dataset where there is an auxiliary knowledge graph which includes input features as nodes.** Similar examples can be found in credit card fraud detection [1] and sensor systems [2] where, for example, every input feature is a value from a massive network of sensors and there is a knowledge graph describing the sensors and their relationships to each other.
>
> **Even computational biology, however, is an extremely large and diverse domain in which we have already constructed the relevant auxiliary knowledge graph for a range of downstream problems and thus believe PLATO is a significant contribution.** The knowledge graph we have already constructed could be directly leveraged for diverse downstream problems including the discovery of novel cell types [3], the prediction of tissue differentiation [4], and viral toxicity in the lung [5]. All of the problems described have tabular datasets in which the input features correspond to nodes in an auxiliary biological knowledge graph and PLATO could thus be used.
>
> References:
>
> [1] Dal Pozzolo, Andrea, et al. "Learned lessons in credit card fraud detection from a practitioner perspective." Expert Systems with Applications 41.10 (2014): 4915-4928.
>
> [2] Harrison, David C., Winston KG Seah, and Ramesh Rayudu. "Rare event detection and propagation in wireless sensor networks." ACM Computing Surveys (CSUR) 48.4 (2016): 1-22.
>
> [3] Haber, Adam L., et al. "A single-cell survey of the small intestinal epithelium." Nature 551.7680 (2017): 333-339.
>
> [4] Dutkowski, Janusz, and Trey Ideker. "Protein networks as logic functions in development and cancer." PLoS computational biology 7.9 (2011): e1002180.
>
> [5] Bajwa, Gagan, et al. "Cutting edge: Critical role of glycolysis in human plasmacytoid dendritic cell antiviral responses." The Journal of Immunology 196.5 (2016): 2004-2009.
>
> ---

---

> > ### Author Response · Authors · 2022-11-14
> > **Response to yX2X (Part 2)**
> >
> > > **4. What is the novelty compared to graph regularization approaches? Did the author compare to graph regularization approaches?**
> >
> > We thank the reviewer for their excellent question!
> >
> > **We add three state-of-the-art graph regularization methods as baselines and demonstrate that PLATO significantly outperforms these new baselines on all six $ d \gg n $ datasets.** Each of the graph regularization methods considers an auxiliary graph on just the input feature nodes with a single edge type. We select GraphNet [1], network-constrained LASSO [2], and network LASSO [3] as these represent state-of-the-art graph regularization approaches for the scenario in which there is an auxiliary graph with input features as nodes. Note that the SConES paper referenced by the reviewer yX2X is an instantiation of Graph Laplacian regularization which is equivalent to GraphNet for our task.
> >
> > | Model                     | MNSCLC          | CM              | PDAC            | BRCA            | CRC             | CH              |
> > |---------------------------|-----------------|-----------------|-----------------|-----------------|-----------------|-----------------|
> > | GraphNet                  | 0.083±0.022     | 0.197±0.018     | 0.132±0.021     | 0.167±0.000     | 0.183±0.011     | 0.048±0.000     |
> > | Network-constrained LASSO | 0.050±0.000     | 0.163±0.020     | 0.124±0.010     | 0.170±0.065     | 0.146±0.009     | 0.048±0.000     |
> > | Network LASSO             | 0.157±0.043     | 0.181±0.052     | 0.110±0.005     | 0.206±0.015     | 0.172±0.055     | 0.048±0.000     |
> > | PLATO                     | **0.272±0.130** | **0.432±0.022** | **0.400±0.021** | **0.583±0.019** | **0.401±0.019** | **0.770±0.003** |
> >
> > **Moreover, PLATO has two major conceptual contributions compared to graph regularization approaches. First, by using a heterogeneous knowledge graph, PLATO can model a broad range of prior information about the input features and the domain generally that is fundamentally missed by graph-regularization approaches.** Graph-regularization methods use a homogeneous graph with input features as nodes to smooth the weights of a predictive model. In the homogeneous graph of a graph regularization, there is a single node type (i.e. an input feature node) and a single edge type (i.e. typically a correlation between input features). By contrast, PLATO uses a **heterogeneous knowledge graph.** The knowledge graph contains input features as nodes and also nodes that represent broader knowledge about the domain. Moreover, the knowledge graph allows for the existence of multiple edge types between any two nodes. The broader domain nodes and existence of multiple edge types enables PLATO to represent a broader range of prior information that is fundamentally missed by graph regularization approaches which consider only feature nodes and one relation type between features. In Table 3 of the manuscript, we demonstrate via an ablation study that the broader domain nodes in the knowledge graph (i.e. the nodes that are not feature nodes) that only PLATO can capture are critical for predictive performance.
> >
> > **Second, PLATO presents a novel mechanism for using an auxiliary graph to constrain the trainable parameters of a model.** Graph regularization approaches typically add a regularization penalty on the loss function which forces weights corresponding to adjacent features to be close to each other. By contrast, PLATO **predicts** the weights of the MLP from prior information about the input feature through a shared neural network with trainable parameters. The prior information is learned from the auxiliary graph as embeddings for the input features. To our knowledge, there are no graph regularization approaches that accomplish regularization via prediction of parameters from an auxiliary graph. PLATO thus provides a novel mechanism for constraining the trainable parameters of a model with an auxiliary graph via parameter prediction.
> >
> > **In response to the reviewer, we have run a new experiment which we will include in the revised version of the manuscript. We will additionally add the discussion of graph regularization models to the Related Work and Appendix. Thank you for the excellent suggestion which makes the paper stronger!**
> >
> > References:
> >
> > [1] Grosenick, Logan, et al. "Interpretable whole-brain prediction analysis with GraphNet." NeuroImage 72 (2013): 304-321.
> >
> > [2] Li, Caiyan, and Hongzhe Li. "Network-constrained regularization and variable selection for analysis of genomic data." Bioinformatics 24.9 (2008): 1175-1182.
> >
> > [3] Hallac, David, Jure Leskovec, and Stephen Boyd. "Network lasso: Clustering and optimization in large graphs." Proceedings of the 21th ACM SIGKDD international conference on knowledge discovery and data mining. 2015.
> >
> > ---

---

> > > ### Author Response · Authors · 2022-11-14
> > > **Response to yX2X (Part 3)**
> > >
> > > > **5. Could the authors more clearly explain how the algorithm was designed?**
> > >
> > > **The PLATO algorithm was designed to accomplish two goals. First, PLATO was designed to reduce the number of trainable parameters (i.e. weights) in the first layer of a MLP by predicting the parameters from the auxiliary knowledge graph. Second, PLATO was designed to allow the parameters in the first layer of the MLP to vary for different input samples, thereby increasing the representational capacity of the MLP.**
> > >
> > > The algorithm itself has three main steps with corresponding motivations.
> > >
> > > 1. [Line 1] First, PLATO pretrains a $c$-dimensional embedding $ \mathbf{M}_j \in \mathbb{R}^c$ for every input feature $ j $. Pre-training occurs via a self-supervision objective on the KG alone (i.e. no tabular data is used). The motivation of pre-training is to capture general prior information about every input feature from the auxiliary KG (Section 3.3.1).
> > > 2. [Lines 2-7] Second, PLATO learns a new embedding $\mathbf{Q}_j \in \mathbb{R}^c $ for every input feature $ j $ by using message-passing on the knowledge graph. In each round of message passing, the embedding of a feature is a weighted combination of its embedding in the previous round and of all its neighbors in the previous round. The weights used in the weighted sum are calculated by a neural network $ \mathcal{A} $ which considers the input feature values $ \mathbf{X}_i$. As a result, the calculated embeddings $ \mathbf{Q}_j $ vary for every input sample $ \mathbf{X}_i$.
> > > The first motivation of message passing is to capture the network principle of homophily, in which adjacent nodes tend to have similar characteristics [1-3]. Message passing allows each input feature embedding to consider the embedding of its neighbors, capturing homophily. The second motivation of message passing is to allow the parameters in the first layer of the MLP to vary for every input sample $\mathbf{X}_i$. Since $\mathbf{Q}_j$ depends on $ \mathbf{X}_i$ and the parameters in the first layer of the MLP are predicted from $ \mathbf{Q}_j$, the parameters in the first layer of the MLP also vary for every $ \mathbf{X}_i$.
> > > 3. [Line 8] Third, PLATO predicts the parameters $ \mathbf{\hat{\Theta}}^{[1]}$ in the first layer of the MLP from each input feature embedding $ \mathbf{Q}_j$. For every input feature $ j $, PLATO uses a small neural network $ \mathcal{P} $ to predict $ \mathbf{\hat{\Theta}}^{[1]}_j$ from $ \mathbf{Q}_j$. $ \mathcal{P} $ and its parameters are shared across all input features and samples, enabling a drastic reduction in parameters compared to the parameters in the first layer of a standard MLP (Section 3.3.3). Since $ \mathbf{Q}_j $ depends on $ \mathbf{X}_i$, the parameters in the first layer of the MLP also vary with $ \mathbf{X}_i$.
> > > 4. [Line 9-10] Finally, PLATO uses the MLP to make a prediction on the input sample $ \mathbf{X}_i$.
> > >
> > > References:
> > >
> > > [1] McPherson, Miller, Lynn Smith-Lovin, and James M. Cook. "Birds of a feather: Homophily in social networks." Annual review of sociology (2001): 415-444.
> > >
> > > [2] Gerber, Elisabeth R., Adam Douglas Henry, and Mark Lubell. "Political homophily and collaboration in regional planning networks." American Journal of Political Science 57.3 (2013): 598-610.
> > >
> > > [3] Zhu, Jiong, et al. "Beyond homophily in graph neural networks: Current limitations and effective designs." Advances in Neural Information Processing Systems 33 (2020): 7793-7804.
> > >
> > > ---
> > >
> > > > **6. Could the authors clarify line 3 of Algorithm 1?**
> > >
> > > **Line 3 of Algorithm 1 explains how the weights are computed for the message passing step of PLATO.** In PLATO’s message passing step, each pretrained feature embeddings $ \mathbf{M}_j \in \mathbb{R}^{c} $ is updated to a feature embedding $ \mathbf{Q}_j \in \mathbb{R}^c$ which depends on the input sample $ \mathbf{X}_i$. In each round of message passing, the embedding of a feature is a weighted combination of its embedding in the previous round and of all its neighbors in the previous round. The weights used in the weighted sum are calculated by a neural network $ \mathcal{A} $ which considers the input feature values $ \mathbf{X}_i$. Line 3 is simply a softmax normalization of the weights computed by neural network $ \mathcal{A} $ across the neighbors of input feature $ j $ in the knowledge graph. Line 3 is closely related to the attention mechanism in Graph Attention Networks [1] which has been demonstrated to be effective in a range of domains [2-3].
> > >
> > > References:
> > >
> > > [1] Veličković, Petar, et al. "Graph attention networks." ICLR (2018).
> > >
> > > [2] Xiong, Zhaoping, et al. "Pushing the boundaries of molecular representation for drug discovery with the graph attention mechanism." Journal of Medicinal Chemistry 63.16 (2019): 8749-8760.
> > >
> > > [3] Shlomi, Jonathan, Peter Battaglia, and Jean-Roch Vlimant. "Graph neural networks in particle physics." Machine Learning: Science and Technology 2.2 (2020): 021001.
> > >
> > > ---

---

> > > > ### Author Response · Authors · 2022-11-14
> > > > **Response to yX2X (Part 4)**
> > > >
> > > > > **7. How are the error bars computed?**
> > > >
> > > > The error bar computation is described in the section “Fair Comparison of PLATO with Baselines.” Specifically, “We report the mean and standard deviation of the Pearson correlation (PearsonR) between $\mathbf{y}$ and $\mathbf{\hat{y}}$ across runs and splits on the test set.”
> > > >
> > > > ---
> > > >
> > > > > **8. PCA is not a supervised method. How can it be used in the experimental tables?**
> > > >
> > > > The PCA row in Table 2 corresponds to PCA followed by linear regression. PCA can thus be used as a relevant baseline for Table 2 and Table 3.
> > > >
> > > > ---
> > > > > **9. Given that no code is available, the reproducibility is limited.**
> > > >
> > > > We will release all code and datasets in the final version of the manuscript as is stated in the Appendix (or by the end of the rebuttal period if we have time). All methodological details are already in the manuscript, making PLATO reproducible. If the reviewer has any further concerns, please let us know!
> > > >
> > > > ---

---

> > > > > ### Author Response · Authors · 2022-11-17
> > > > > **Thank you!**
> > > > >
> > > > > We thank the reviewer for their time and thoughtful questions! If the reviewer has any further concerns, please let us know! If not, we would appreciate the reviewer considering raising their score. Thank you!

---

> > > > > > ### Author Response · Authors · 2022-12-13
> > > > > > **Final commentary**
> > > > > >
> > > > > > If the reviewer has had the chance to read our response, we would appreciate any final commentary! If there are no further concerns, we would appreciate the reviewer considering raising their score. Thank you!

---

> > > > > > > ### Comment · Reviewer_yX2X · 2022-12-14
> > > > > > > **More baselines improved the manuscript, but the generality still not established.**
> > > > > > >
> > > > > > > I thank the authors nudging me into replying their very lengthy responses. I do admit that I have not do a very good job of engaging in the discussion, because I am still not convinced that this work passes the very high bar of ICLR (which does not mean that it is a bad work per se). I would gladly raise my rating to 4, but the interface does not allow this.
> > > > > > >
> > > > > > > With regards to the other applications, it would have been good to demonstrate the value of the proposed method as right now it requires a leap of faith to believe that it is general. For instance, credit-card fraud detection is not in high-dimensional settings, as far as I am concerned.
> > > > > > >
> > > > > > > I thank the authors for adding the comparison to graph regularization approaches. I believe that it makes the manuscript more convincing and will help it reach a broader impact. It does not seem that the authors added these figures to the manuscript, which is a pity.
> > > > > > >
> > > > > > > I thank the authors for explaining the algorithm, how my questioning is on the process of designing this algorithm: it seems that there are many moving parts, many things that could have been done different (for instance, there are multiple ways of normalizing an attention layer). One wonders if the authors conceived the details of the algorithm by intuition or whether there was an empirical process involved. In the case of an empirical process, a question is then how it was controlled for overfitting.

---

### Official Review · Reviewer_rHdL · 2022-10-25

**Confidence:** 4
**Correctness:** 4
**Technical Novelty And Significance:** 3
**Empirical Novelty And Significance:** 3
**Recommendation:** 5

**Clarity, Quality, Novelty And Reproducibility:**

The paper is generally clear, but the method is a bit complicated. The authors may add the size of matrices in the algorithms.


**Strength And Weaknesses:**

To deal with the case that the dimensionality of the data is very larger than the sample size, the authors introduce an additional knowledge graph as a prior of the feature relationship. Then the weights of the MLP are better determined.

There are some suggestions for the paper:
1. Why the weights of the first layer of the MLP is more important than the others?
2. Do the results depend on the quality of the knowledge graph? How to compare with other methods in a fair manner since an auxiliary knowledge graph is introduced.
3. Pre-train the node embedding in Eq. 2 is not with a self-supervised node embedding method is not clear enough. How will the pretraining method influences the final results?
4. The layer-wise predicted embeddings are concatenated in line 9 in the algorithm. It works like an ensemble. How much does the concatenation step improve in the whole method?
5. Please consider comparing more tabular data methods, such as FT-Transformer based on some dimensionality reduction methods.
6. It seems the column names are required. Does it limit the field of the method?

**Summary Of The Paper:**

To deal with the problem when tabular data has high dimensionality, the paper introduces a knowledge graph to help determine the weights of an MLP. The proposed PLATO method achieves good results on various datasets.


**Summary Of The Review:**

The main idea of the paper is reasonable. An auxiliary knowledge graph will introduce more information especially since there are many features. The method is clear and the results are good. Please check the weakness part for some questions.

---

> ### Author Response · Authors · 2022-11-06
> **Tabular baselines are best-performing from prior benchmarks**
>
> >**5. Please consider comparing more tabular data methods, such as FT-Transformer based on some dimensionality reduction methods.**
>
> We already implemented FT-Transformer but it did not apply to the PLATO datasets. FT-Transformer experiences out of memory issues for datasets with a high number of features (as we noted in the “Baselines” section of “Experiments”). In PLATO, datasets have between 12,693 and 19,902 features. FT-Transformer was designed for datasets with far fewer features. The datasets in the FT-Transformer paper have 6-2000 features, with most datasets having 6-136 features, multiple orders of magnitude fewer features than the PLATO datasets.
>
> More generally, we systematically selected tabular data baselines by picking the best published methods in recent tabular benchmark papers [1-2]. We selected NODE, TabTransformer, TabNet, and FT-Transformer as these were the best performing. Further, note that deep tabular methods are not designed for the d >> n setting. Correspondingly, they generally perform poorly on the PLATO datasets (Table 1). The best performing baselines for the PLATO datasets on all but the MNSCLC dataset are classical approaches (Ridge Regression, LASSO, XGBoost). TabTransformer does the best on MNSCLC but performs poorly on the other datasets.
>
> Please let us know if this adequately resolves the reviewer’s concern!
>
> [1] Grinsztajn, Leo, Edouard Oyallon, and Gael Varoquaux. "Why do tree-based models still outperform deep learning on typical tabular data?." NeurIPS Datasets and Benchmarks Track. 2022.
>
> [2] Yury Gorishniy, Ivan Rubachev, Valentin Khrulkov, and Artem Babenko. Revisiting deep learning models for tabular data. NeurIPS, 34, 2021.

---

> ### Author Response · Authors · 2022-11-14
> **Response to rHdL (Part 1)**
>
> > **The proposed PLATO method achieves good results on various datasets... The main idea of the paper is reasonable. An auxiliary knowledge graph will introduce more information especially since there are many features. The method is clear and the results are good.**
>
> We thank the reviewer for pointing out the numerous strengths of the PLATO methodology and results! We address the reviewers' questions, which strengthen the manuscript, below!
>
> > **1. Why does PLATO focus on the first layer of weights in the MLP? Is that layer more important than the others?**
>
> **PLATO focuses on weights in the first layer of the MLP because it is where the largest number of trainable weights are in the entire MLP when $d \gg n$ (i.e. the dataset has a large number of features $d$ and a small number of samples $n$).** The first layer of MLP weights has $dh$ trainable weights where $h$ is the number of hidden units in the first layer of the MLP. When $d \gg n$, the remainder of the MLP can be designed to be shallow (i.e. few layers) and narrow (i.e. few hidden units per layer) to reduce the number of additional trainable weights. However, the $dh$ trainable weights in the first layer are necessarily large for datasets with a large number of features like those in PLATO and predicting these weights from auxiliary information can thus significantly reduce the number of trainable weights in the MLP (Methods 3.3.3).
>
> Moreover, PLATO predicts the trainable weights in the first layer of the MLP from prior, auxiliary information about each input feature. In the first layer of the MLP, each trainable weight is associated with an input feature (Figure 1e). As a result, we can predict the weight from prior information about the input feature. However, weights in subsequent layers of a MLP are not associated with a specific input feature and such a prediction thus cannot occur.
>
> ---
>
> > **2a. Do PLATO’s results depend on the quality of the knowledge graph?**
>
> We thank the reviewer for their constructive question! In response, we have conducted a **new experiment** to showcase how PLATO’s performance depends on the quality of the knowledge graph.
>
>
> **PLATO’s performance does, in general, depend on the coverage and quality of the knowledge graph.** Necessarily, all knowledge graphs are incomplete since there is additional knowledge yet to be discovered. To handle the incompleteness of the knowledge graph, PLATO uses low-dimensional embeddings that are known to be robust to missing information [1]. **We conduct a new experiment to further demonstrate PLATO’s robustness to missing information in the knowledge graph.** We randomly remove edges from the knowledge graph and measure PLATO’s performance on the BRCA dataset. We observe that with only 50% of the knowledge graph’s edges, PLATO still has 71% of the performance as PLATO with 100% of the knowledge graph’s edges (0.412±0.011 vs. 0.583±0.019). In general, knowledge graphs are incomplete but PLATO’s use of low-dimensional embeddings enables PLATO to capture useful prior information.
>
> | **Incompleteness of KG** | **PearsonR (Test) on BRCA dataset** |
> |--------------------------|-------------------------------------|
> | 50%                      | 0.412±0.011                         |
> | 70%                      | 0.537±0.044                         |
> | 90%                      | 0.570±0.017                         |
> | 100%                     | 0.583±0.019                         |
>
>
> References:
>
> [1] Trouillon, Théo, et al. "Complex embeddings for simple link prediction." International Conference on Machine Learning. PMLR, 2016.
>
> ---

---

> > ### Author Response · Authors · 2022-11-14
> > **Response to rHdL (Part 2)**
> >
> > > **2b. How can PLATO be compared with other methods since PLATO also uses an auxiliary knowledge graph?**
> >
> > We thank the reviewer for their question! We clarify PLATO’s contribution and run a new experiment to show that PLATO outperforms new baselines that also consider an auxiliary graph.
> >
> > **[Conceptual clarification]** PLATO is a deep learning method for tabular data in which there is an auxiliary, heterogeneous knowledge graph (KG) with input features as nodes. To the best of our knowledge, there are not other methods for this novel tabular problem setting. In particular, the KG allows for both input features as nodes as well as other nodes representing broader information. The KG also allows for different relation types among nodes.
> >
> > **[New experiment]** During the rebuttal period, the closest related work we were able to find involves graph regularization. Graph-regularization methods use a homogeneous graph with only input features as nodes and one relation type to smooth the weights of a predictive model. We add three state-of-the-art graph regularization methods as baselines, run a new experiment, and demonstrate that PLATO significantly outperforms these new baselines on all six $ d \gg n $ datasets. Each of the graph regularization methods considers an auxiliary graph on just the input feature nodes with a single edge type. We select GraphNet [1], network-constrained LASSO [2], and network LASSO [3] as these represent state-of-the-art graph regularization approaches for the scenario in which there is an auxiliary graph with input features as nodes.
> >
> > | Model                     | MNSCLC          | CM              | PDAC            | BRCA            | CRC             | CH              |
> > |---------------------------|-----------------|-----------------|-----------------|-----------------|-----------------|-----------------|
> > | GraphNet                  | 0.083±0.022     | 0.197±0.018     | 0.132±0.021     | 0.167±0.000     | 0.183±0.011     | 0.048±0.000     |
> > | Network-constrained LASSO | 0.050±0.000     | 0.163±0.020     | 0.124±0.010     | 0.170±0.065     | 0.146±0.009     | 0.048±0.000     |
> > | Network LASSO             | 0.157±0.043     | 0.181±0.052     | 0.110±0.005     | 0.206±0.015     | 0.172±0.055     | 0.048±0.000     |
> > | PLATO                     | **0.272±0.130** | **0.432±0.022** | **0.400±0.021** | **0.583±0.019** | **0.401±0.019** | **0.770±0.003** |
> >
> > The conceptual contributions of PLATO over graph regularization approaches are further elucidated in our response to Reviewer yX2X.
> >
> > References:
> >
> > [1] Grosenick, Logan, et al. "Interpretable whole-brain prediction analysis with GraphNet." NeuroImage 72 (2013): 304-321.
> >
> > [2] Li, Caiyan, and Hongzhe Li. "Network-constrained regularization and variable selection for analysis of genomic data." Bioinformatics 24.9 (2008): 1175-1182.
> >
> > [3] Hallac, David, Jure Leskovec, and Stephen Boyd. "Network lasso: Clustering and optimization in large graphs." Proceedings of the 21th ACM SIGKDD international conference on knowledge discovery and data mining. 2015.
> >
> > ---

---

> > > ### Author Response · Authors · 2022-11-14
> > > **Response to rHdL (Part 3)**
> > >
> > > > **3. How are the node embeddings pre-trained in Eq. 2 via self-supervision on the knowledge graph? How does the pre-training method influences the final results?**
> > >
> > > **[Conceptual clarification]** PLATO pre-trains node embeddings by using ComplEx, a self-supervised node embedding method for knowledge graphs [1]. ComplEx learns an embedding for every node in the knowledge graph by classifying whether a proposed edge (i.e. a <head, relation, tail> tuple) exists in the knowledge graph or not. ComplEx proposes positive edges (i.e. edges that exist in the knowledge graph) and negative edges (i.e. edges that do not exist in the knowledge graph and are constructed by corrupting positive edges) by sampling. The edges proposed by ComplEx consider nodes across the entire graph, enabling PLATO to capture a broad range of prior information about each input feature when the embedding for the corresponding node is learned. In the revised version of the manuscript, we will add a section in the Appendix to detail how ComplEx works more thoroughly.
> > >
> > > More generally, PLATO makes no assumption about what type of self-supervised node embedding method is used. The self-supervised embedding is simply a module to pre-train feature embeddings which are then passed to the message passing and parameter prediction modules of PLATO.
> > >
> > > **[New experiment] In response to the reviewer’s comment, we conduct a new experiment and demonstrate that PLATO’s performance is robust across three self-supervised node embedding methods.** We compare the ComplEx embedding model described above with TransE [2] and DistMult [3], two alternate embedding approaches for heterogeneous knowledge graphs. ComplEx exhibits the best performance, though PLATO’s performance is similar across the KG embedding methods.
> > >
> > > | **KG Embedding Method** | **PearsonR (Test) on BRCA dataset** |
> > > |-------------------------|-------------------------------------|
> > > | TransE                  | 0.582±0.025                         |
> > > | DistMult                | 0.575±0.011                         |
> > > | ComplEx                 | 0.583±0.019                         |
> > >
> > > References:
> > >
> > > [1] Trouillon, Théo, et al. "Complex embeddings for simple link prediction." International conference on machine learning. PMLR, 2016.
> > >
> > > [2] Wang, Zhen, et al. "Knowledge graph embedding by translating on hyperplanes." Proceedings of the AAAI conference on artificial intelligence. Vol. 28. No. 1. 2014.
> > >
> > > [3] Yang, Bishan, et al. "Embedding entities and relations for learning and inference in knowledge bases." ICLR (2014).
> > >
> > > ---
> > > > **4. In line 9 of the algorithm, the layer-wise predicted embeddings are concatenated. It works like an ensemble. How much does the concatenation step improve the whole method?**
> > >
> > > We appreciate the opportunity to clarify!
> > >
> > > **PLATO does not use ensembling. PLATO predicts the parameters in the first layer of a MLP and trains the parameters in the remaining layer of the MLP normally.** In line 9 of Algorithm, the predicted parameters $\mathbf{\hat{\Theta}}^{[1]}$ that PLATO has predicted are concatenated with the parameters in the rest of the MLP’s layers $\mathbf{\Theta}^{[2]}, \mathbf{\Theta}^{[3]}, \ldots$ which are trained normally.
> > >
> > > **If the parameters in the first layer of the MLP are trained normally rather than predicted by PLATO, the performance drops significantly.** Below, we compare the performance of PLATO’s MLP with a MLP in which the parameters in the first layer are learned normally. All models are hyperparameter-tuned across a range of configurations to ensure a fair comparison (details in Experiments section and Appendix A). **As a result, PLATO’s prediction of the parameters in the first layer of the MLP is a critical methodological component of its effectiveness.**
> > >
> > > |       | MNSCLC          | CM              | PDAC            | BRCA            | CRC             | CH              |
> > > |-------|-----------------|-----------------|-----------------|-----------------|-----------------|-----------------|
> > > | MLP   | 0.128 +/- 0.126 | 0.322 +/- 0.043 | 0.289 +/- 0.047 | 0.240 +/- 0.067 | 0.355 +/- 0.022 | 0.044 +/- 0.039 |
> > > | PLATO | 0.272 +/- 0.130 | 0.435 +/- 0.022 | 0.400 +/- 0.021 | 0.583 +/- 0.019 | 0.401 +/- 0.019 | 0.770 +/- 0.003 |
> > >
> > > ---
> > >
> > > > **5. Please consider comparing more tabular data methods, such as FT-Transformer based on some dimensionality reduction methods.**
> > >
> > > We address this question in the previous comment we posted in response to this reviewer rHdL titled "Tabular baselines are best-performing from prior benchmarks"!
> > >
> > > ---

---

> > > > ### Author Response · Authors · 2022-11-14
> > > > **Response to rHdL (Part 4)**
> > > >
> > > > > **6. It seems the column names are required. Does it limit the field of the method?**
> > > >
> > > > **The column names (i.e. the name of a given input feature) are not required as long as the column (i.e. input feature) corresponds to a node in an auxiliary knowledge graph.** The auxiliary knowledge graph will capture the relationship between different columns as well as other prior information known about each column. As a result, there is no limitation to the field of PLATO’s application. We further specify the problem setting in Methods 3.1.
> > > >
> > > > ---
> > > >
> > > > > **7. To improve clarity, the authors may add the size of the matrices in Algorithm 1.**
> > > >
> > > > We thank the reviewer for their excellent suggestion. In the revised version of the manuscript, we will update Algorithm 1 to include the size of all matrices!
> > > >
> > > > ---

---

> > > > > ### Author Response · Authors · 2022-11-17
> > > > > **Thank you!**
> > > > >
> > > > > We thank the reviewer for their time and thoughtful questions! If the reviewer has any further concerns, please let us know! If not, we would appreciate the reviewer considering raising their score. Thank you!

---

> > > > > > ### Author Response · Authors · 2022-12-13
> > > > > > **Final commentary**
> > > > > >
> > > > > > If the reviewer has had the chance to read our response, we would appreciate any final commentary! If there are no further concerns, we would appreciate the reviewer considering raising their score. Thank you!

---

### Official Review · Reviewer_AKNa · 2022-10-30

**Confidence:** 4
**Correctness:** 4
**Technical Novelty And Significance:** 4
**Empirical Novelty And Significance:** 3
**Recommendation:** 8

**Clarity, Quality, Novelty And Reproducibility:**

Clearly written paper. Experimental results are rigorous and following best practices including hyper parameter optimization and results reported over a large number of runs. As far as I am aware, this is a novel approach to solving for an underserved set of applications common in biology and physical sciences where d>>n

**Strength And Weaknesses:**

Strengths
1. Solves for an important set of applications that are underserved in deep learning focused research geared towards large datasets.
2. Elegant way of incorporating prior information via a knowledge graph (KG). While not all applications have associated knowledge graphs, a sizeable set do. KG is a distillation of several studies, and to be able to incorporate effectively is significant.

3. Experimental results are rigorous. On multiple datasets and compared with multiplied methods (both statistical and deep-learning based) Established protocols for tabular dataset are followed.
4. The weights of the first layer of MLP are learned in 3 steps described above. An ablation study shows the importances of each step showing the added value of a message passing routine and also of adding knowledge about features not present in the datasets.
5. Clearly describes architecture, datasets, and results.

Weaknesses:
Ablation study is presented only on a single BRCA dataset. It would be valuable for readers to know that findings hold across datasets.

The intro and related work sections both spend significant ink on related work, including describing works like graph-based prediction that are only tangentially related. Instead, the intro can incorporate a summary of the contributions in the paper. Currently, the reader has to read all through methods and results to understand the significance of the contributions. Examples of items to include earlier are how message passing the graph structure of KG is used effectively to incorporate additional info. This is a significant result not sufficiently highlighted.

The current paper is not incomplete without these, but some suggestions for future work:

Study impact of missingness in KG. Is it possible to support datasets where some subset of features are not described in KG. The paper currently assumes every feature in dataset is a node in the KG.

A study on model's sample complexity e.g., we broadly cast this problem as d>>n. e.g., can we afford to drop some rows from the dataset and the models achieve the same performance.. What is the point of diminishing returns. It would be interesting to see how few examples the model can get away with.

**Summary Of The Paper:**

Authors define PLATO, a model for underserved scenario of short and fat datasets (d>>n). PLATO works when we have an auxiliary knowledge graph describing the d features and the relations between them. Tabular datasets with d>>n are common in physical sciences and biology where data is collected through expensive experiments.

Plato is an MLP where the first layer is trained by leveraging the knowledge graph as follows:
1. A c-dimensional embedding M_j is learned for each feature j
2. A message passing algorithm that uses the knowledge graph is used to learn another embedding Q_j for each feature j based on its neighbors. In each round the embedding of a feature is a weighted combination of its embedding in the previous round and all its neighbors in the previous round.
3. A neural network outputs P the weights of the first layer.

Experimental results are presented on multiple datasets and compared with multiplied methods (both statistical and deep-learning based). Some experimental results are presented to explain the motivation around using a message passing protocol and large knowledge graphs describing features even if not all are in dataset.

**Summary Of The Review:**

Overall, the authors present a novel approach to supervised learning tasks on datasets where d>>n. Given the novelty of the approach, significance of the setting for biological and physical sciences and elegance of the solution, I vote accept.

The major weaknesses of the paper are largely open questions that can be addressed in a future iteration of the paper. The paper remains of interest to the community  at this iteration.

---

> ### Author Response · Authors · 2022-11-19
> **Response to AKNa (Part 1)**
>
> > **Overall response**
>
> We thank the reviewer for their **excellent review**! The reviewer points out many strengths of the paper, synthesized in the sentence: **“Given the novelty of the approach, significance of the setting for biological and physical sciences, and elegance of the solution, I vote accept.”**
>
> The reviewer further points out strengths including the **“rigorous experimental results”** which **“[follow] best practices,”** the **“ablation [studies]”** which **“show the importance of each step”** of the method, and the **“clear”** description of **“architecture, datasets, and results.”** The reviewer concludes that PLATO is **“of interest to the community at this iteration.”**
>
> ---
>
> > **1. The authors' ablation studies (Table 2, 3) are presented on the BRCA dataset. The authors should consider repeating these studies on additional datasets.**
>
> We thank the reviewer for their comment and agree! We are currently **repeating the ablation studies in Tables 2 and 3 on an additional $d \gg n$ dataset in the main table.** We will **add these to the final version of the manuscript**!
>
> ---
>
> > **2. The authors should include a summary of the paper’s contributions at the end of the Introduction. Space can be saved by trimming the related works section, particularly discussion of works like graph prediction which are only tangentially related.**
>
> **We thank the reviewer for their excellent suggestion and agree!** In the revised version of the manuscript, **we will add the following to the Introduction and trim the related works section.**
>
> PLATO's key contributions are:
>
> 1. The development of a deep learning model that makes effective predictions on tabular datasets with $d \gg n$ by integrating an auxiliary knowledge graph with input features as nodes. This model includes:
>
> a. A message-passing architecture which learns an embedding for every input feature in the auxiliary KG with minimal trainable parameters while sharing information across related input features.
>
> b. A parameter-prediction method for the first layer of weights in the MLP which drastically reduces the number of trainable parameters compared to the first layer of a standard MLP and allows these weights to vary for every input sample, increasing the MLP's representational capacity.
>
> 2. The use of a single auxiliary knowledge graph as a unified source of prior knowledge for diverse tabular datasets with distinct input features and $d \gg n$.
>
> ---

---

> > ### Author Response · Authors · 2022-11-19
> > **Response to AKNa (Part 2)**
> >
> > > **3. The current paper is not incomplete without these, but here are suggestions for future work. Can PLATO support datasets where some subset of the input features are not nodes in the KG? PLATO currently assumes every input feature in the dataset is a node in the KG.**
> >
> > We thank the reviewer for their excellent question! **Expanding PLATO to settings where only a subset of the input features are present as nodes in the auxiliary KG is an exciting avenue for future work.** We suggest two potential approaches below.
> >
> > First for **context**, recall that PLATO is a MLP in which the weights of the first layer are predicted from an auxiliary KG. In the standard PLATO setting, the auxiliary KG includes every input feature as a node. Since every weight in the first layer of a MLP corresponds to an input feature, PLATO predicts the weights corresponding to an input feature from prior information about that input feature that is present in the auxiliary KG.
> >
> > To expand to a problem setting in which a subset of features are not nodes in the auxiliary KG, we suggest two potential approaches for future work.
> >
> > **First, PLATO could be used to predict only a subset of the weights in the first layer of the MLP.** PLATO would predict only the weights that are associated with input features that are nodes in the auxiliary KG. PLATO would learn the remaining weights in the first layer of the MLP normally. In this scenario, PLATO could still achieve a large reduction in the number of trainable parameters in the first layer of the MLP, as long as a large majority of the input features are nodes in the auxiliary KG (Section 3.3.3).
> >
> > **Second, PLATO could be used to predict all weights in the first layer of the MLP by considering the correlation between input feature values.** To predict the weights associated with a given input feature, PLATO must learn an embedding for that input feature. Let $A$ be the set of input features that are nodes in the KG and $B$ be the set of the remaining input features. To learn an embedding for the input features that are nodes in the KG (i.e. for all input features $j \in A$), PLATO would follow its standard methodology to learn $\mathbf{Q}_j \in \mathbb{R}^c$ (Section 3.3.1, 3.3.2). Then, for every input feature that is not a node in the auxiliary KG, PLATO would construct an embedding from a weighted sum that considers the correlations between the input feature values in the tabular data. For a feature not present in the auxiliary KG (i.e. for $j \in B$), PLATO would construct an embedding according to $\mathbf{Q}_j = \sum _{k \in A} \mathrm{corr}(\mathbf{X} ^{T} _{j}, \mathbf{X} ^{T} _{k}) \mathbf{Q}_k$. $\mathrm{corr}(\mathbf{X} ^{T} _{j}, \mathbf{X} ^{T} _{k})$ is a scalar correlation coefficient between the values that feature $j$ takes on across the training samples and the values that feature $k$ takes on across the training samples. With an embedding for every input feature, PLATO could then predict the weights in the first layer of the MLP by following its standard methodology (Section 3.3.3).
> >
> > We thank the reviewer for the excellent question! In the revised version of the manuscript, we will add PLATO's potential application to datasets with some features not in the KG as a direction for future work!
> >
> > ---
> >
> > > **4. How does PLATO’s performance vary with the raw number of labeled input samples? How many samples can PLATO drop from the input tabular dataset and maintain its predictive performance?**
> >
> > We thank the reviewer for their excellent suggestion! In the revised version of the manuscript, we will consider adding an experiment which assesses PLATO’s performance on a $d \gg n$ dataset as the number of training samples are downsampled. We agree that this is an exciting direction for future work!
> >
> > ---

---

> > > ### Author Response · Authors · 2022-12-13
> > > **Final commentary?**
> > >
> > > If the reviewer has had the chance to read our response, we would appreciate any final commentary! Thank you!

---

### Author Response · Authors · 2022-11-19
**Overall response to reviewers**

> **Overall Response to Reviewers**

We thank the reviewers for their thoughtful comments! PLATO is a deep learning method for tabular datasets with far more $d$-features than $n$-samples (i.e. $d \gg n$). PLATO's key insight is to use an auxiliary knowledge graph (KG) with input features as nodes to predict the parameters in the first layer of a MLP, thereby drastically reducing the number of trainable parameters and enabling effective predictions.

The reviewers appreciated the **"novel"** and **“elegant** way of incorporating prior information via a KG", the **"rigorous experimental results,"** and the **"important"** yet **"underserved"** problem setting which is **"common" and "significant"** in "physical sciences and biology."

The reviewer's questions encouraged us to compare PLATO with additional graph regularization baselines, run additional experiments to further characterize PLATO's performance, and clarify specific steps of our methodology. We thank the reviewers for their insightful questions which have strengthened the manuscript! Our response includes:

1. **New experiments** that compares PLATO with three state-of-the-art graph regularization baselines. We show that **PLATO outperforms new graph regularization baselines on all six $d \gg n$ datasets.** We also conceptually describe the contributions of PLATO vs. these and other graph regularization approaches.

2. Additional experiments characterizing PLATO's performance, including:

2a. A **new experiment** that **characterizes PLATO's robustness to missing information in the KG.**

2b. A **new experiment** demonstrating **PLATO’s robustness across three distinct pre-training methodologies** for node embeddings.

3. **Clarification of specific steps** of the PLATO algorithm and further explanation of the mathematical justifications for its design. Many of these clarifications draw from Reviewer AKNa's excellent summary of our method. Incorporating these clarifications in the revised version of the manuscript will further improve its readability!

Overall, PLATO beats thirteen baselines spanning statistical approaches, deep learning, and graph regularization on six $d \gg n$ datasets, achieving performance improvements of up to +10.19%. **We believe the "novelty" of PLATO's methodology and its "significant" application to "underserved" and "common" datasets in "physical sciences and biology" make it “of interest to the community”.** Thank you for your consideration!

---

### Comment · Area_Chair_NCqN · 2022-11-26
**Following up on authors’ response and discussion**

Dear Reviewers,

Thank you very much again for performing this extremely valuable service to the ICLR community.

Please check the authors’ response and leave comments if you have not done it.

Best,

AC

---

### Decision · Program_Chairs · 2023-01-20

**Decision:**

Reject

**Justification For Why Not Higher Score:**

While the authors provided response to address the concerns of reviewers, it was not enough to convince the negative reviewers. Hence, given the current status, AC recommends reject.

**Justification For Why Not Lower Score:**

N/A

**Metareview: Summary, Strengths And Weaknesses:**

This paper suggests a technique for tabular datasets with high-dimensional features but limited number of samples. Specifically, the authors suggest to utilize an auxiliary knowledge graph to predict the first layer parameters of the MLP, effectively reducing the number of trainable parameters. Three reviewers are negative and one reviewer is positive. The main concerns of negative reviewers are unclear motivation for several design choices, limited applicability from assuming a knowledge base and the lack of novelty compared to graph regularization work approaches.